# Eye movements track prioritized auditory features in selective attention to natural speech

Quirin Gehmacher [1] ✉, Juliane Schubert[1], Fabian Schmidt [1], Thomas Hartmann [1], Patrick Reisinger [1], Sebastian Rösch [2], Konrad Schwarz[3], Tzvetan Popov [4,5], Maria Chait [6] & Nathan Weisz[1,7]

Over the last decades, cognitive neuroscience has identified a distributed set of brain regions that are critical for attention. Strong anatomical overlap with brain regions critical for oculomotor processes suggests a joint network for attention and eye movements. However, the role of this shared network in complex, naturalistic environments remains understudied. Here, we investigated eye movements in relation to (un)attended sentences of natural speech. Combining simultaneously recorded eye tracking and magnetoencephalographic data with temporal response functions, we show that gaze tracks attended speech, a phenomenon we termed ocular speech tracking. Ocular speech tracking even differentiates a target from a distractor in a multi-speaker context and is further related to intelligibility. Moreover, we provide evidence for its contribution to neural differences in speech processing, emphasizing the necessity to consider oculomotor activity in future research and in the interpretation of neural differences in auditory cognition.

The brain is highly efficient in processing a vast amount of information in complex environments, thereby enabling adaptive behavior. A key principle of adaptive behavior is the goal-directed prioritization and selection of relevant events or objects by attention. From a neurobiological perspective, a distributed attention network extending from relevant sensory cortices to temporal, parietal, and frontal regions[1–4] shows a strong anatomical overlap with brain regions critical for oculomotor processes, suggesting a joint network for attention and eye movements[5–7].

Just as eye movements are necessary for the goal-directed exploration of the visual field, gathering and evaluating additional information via omnidirectional hearing is an inevitable requirement for action preparation and adaptive behavior. Studies on monkeys and cats suggest a midbrain level hub of the inferior colliculus (IC, an obligatory station of the ascending auditory pathway) and superior colliculus (SC, which controls ocular dynamics) to integrate sounds and visual scenes via eye movements[8–10]. This circuit has recently been extended to the auditory periphery in humans[11,12]. Accordingly, several studies in humans point towards interactions between eye movements and auditory cognition in sound localization[13], spatial discrimination[14], and spatial attention[15] with lateralized engagement of the posterior parietal cortex in unison with lateralized gaze direction[16]. However, the role of a shared network of auditory attention and eye movements in more complex, naturalistic listening situations remains largely unknown.

Speech represents a key component of social communication that requires a highly selective allocation of spatial, temporal, and feature-based attention. In a mixture of spatially distributed speakers (i.e. "cocktail party scenario"), orienting the eyes towards the target source seems to increase intelligibility[17], and eye blinks are more likely to

[1]Paris-Lodron-University of Salzburg, Department of Psychology, Centre for Cognitive Neuroscience, Salzburg, Austria. [2]Department of Otorhinolaryngology, Head and Neck Surgery, Paracelsus Medical University Salzburg, 5020 Salzburg, Austria. [3]MED-EL GmbH, 6020 Innsbruck, Austria. [4]Methods of Plasticity Research, Department of Psychology, University of Zurich, CH-8050 Zurich, Switzerland. [5]Department of Psychology, University of Konstanz, DE- 78464 Konstanz, Germany. [6]Ear Institute, University College London, London, UK. [7]Neuroscience Institute, Christian Doppler University Hospital, Paracelsus Medical University, Salzburg, Austria. ✉e-mail: quirin.gehmacher@plus.ac.at

occur during pauses in target speech compared to distractor speech[18]. In addition, Jin and colleagues[19] showed that blink-related eye activity aligned with higher-order syntactic structures of temporally predictable, artificial speech (i.e. monosyllabic words), similar to neural activity. Their results suggest a global neural entrainment across sensory and (oculo)motor areas which implements temporal attention, supporting ideas that the motor system is actively engaged in speech perception[20,21]. Taken together, the evidence strongly suggests an engagement of the oculomotor system in auditory selective attention even in more complex scenes involving speech, and this engagement also seems to support adaptive behavior. However, several important questions that are essential for a comprehensive understanding of a joint network of auditory attention and eye movements remain unanswered:

Firstly, it is unknown whether eye movements (aside from blinking) continuously track ongoing acoustics of speech, especially in naturalistic scenarios where features often overlap in a mixture of target and distracting sources. This ocular speech tracking could be sensitive to selective attention, for example by gaze reorientation in concordance with relevant structures of attended speech streams. Crucially, the absence of any spatial cues or discriminability could additionally provide valuable information on the underlying principles of oculomotor action in auditory selective attention. Secondly, it is unknown whether ocular speech tracking is related to adaptive behavior, as quantification of important markers like intelligibility or effort is, to date, lacking. Thirdly, the contribution of eye movements to neural processes and underlying computations in selective attention to speech has been overlooked completely. In their aforementioned study, Popov et al.[16] indicated that a partial contribution of goal-driven oculomotor activity to typical cognitive effects in spatial auditory attention was retained even after removing scalp signal variance (e.g., by means of independent component analysis, ICA) related to ocular muscle activity. Based on their findings, it is important to address this possible contribution when evaluating neural responses in selective attention to speech.

To answer these questions, we analyzed simultaneously recorded eye tracking and magnetoencephalographic (MEG) data from participants listening to short sentences of natural speech at the phantom center. Critically, we manipulated attention within and across modalities such that sentences were presented as distractors (Condition 1), as targets (Condition 2), or as a mixture of target and distractor in a multi-speaker scenario (Condition 3). Using temporal response functions (TRF[22,23]), we show that attended features of speech (i.e. envelope and acoustic onsets) are, in fact, tracked by eye movements. Crucially, ocular speech tracking is stronger for a target compared to a distractor speaker in the multi-speaker condition. Furthermore, this ocular tracking of speech features is related to intelligibility. Finally, using a mediation analysis approach, we show that eye movements and selective attention to speech share neural mechanisms over left temporoparietal sensors, suggesting a contribution of ocular speech tracking to brain regions typically involved in speech processing (or vice versa). Taken together, these results extend previous evidence for a joint network of auditory selective attention and eye movements in complex naturalistic environments. They provide implications on adaptive behavior and moreover suggest a shared contribution of oculomotor and neural activity in auditory selective attention that should be taken into consideration by future research on auditory cognition.

## Results

Participants listened to short sentences of natural speech in different conditions of selective attention. Sentences featured either a single speaker as a distractor to visual attention, a single speaker as the target of the auditory modality, a target in a multi-speaker condition, and consequently a distractor in a multi-speaker condition (see Fig. 1).

Using TRFs, a regularized linear regression approach, we evaluated ocular speech tracking based on the model's ability to predict held-out eye movement data in a nested cross-validation procedure (see Fig. 2). Eye tracking and MEG data were simultaneously recorded. First, we investigated spatial gaze characteristics during trials. Importantly, densities confirm that participants kept their gaze on the visual stimulus (i.e. Gabor patch) at the center of the screen (see Fig. 3a). Additionally, we contrasted Conditions 2 & 3 (attend speech) against Condition 1 (attend Gabor) using a cluster-based permutation test (see Fig. 3b). For both contrasts, we observed a slight shift of gaze to the top-right whenever the auditory modality is attended. Contrasting Condition 2 (attend single speaker) against Condition 1 revealed one positive cluster ($t(29) = 7.72$, $p < 0.001$, Cohen's $d = 1.41$), and one negative cluster ($t(29) = −11.34$, $p < 0.001$, Cohen's $d = −2.07$). Similarly, the cluster-based permutation test revealed one positive cluster ($t(29) = 6.26$, $p < 0.001$, Cohen's $d = 1.14$), and one negative cluster ($t(29) = −8.69$, $p < 0.001$, Cohen's $d = −1.59$) for the contrast Condition 3 (attend a target in a multi-speaker condition) vs. Condition 1.

### Eye movements track prioritized acoustic features of natural speech

We then answered the question as to whether eye movements track natural speech, and how this tracking is modulated by selective attention and subsequently provided evidence that ocular speech tracking prioritizes acoustic features of a target speaker, even in the presence of a simultaneously presented distractor. Bayesian multilevel models with Fisher z-transformed prediction accuracies of ocular speech tracking as dependent variables (z') revealed compelling evidence that eye movements only track the envelope of a single speaker when it was presented as the target of attention ($\beta = 0.032$, 94%HDI = [0.022, 0.041]), not when it served as a distractor to the visual modality ($\beta = 0.006$, 94%HDI = [−0.004, 0.015]). We observed a similar effect when using acoustic onsets as a predictor, indicating substantial evidence for ocular tracking of the target ($\beta = 0.040$, 94%HDI = [0.030, 0.050]) but not the distractor sentences ($\beta = 0.004$, 94%HDI = [−0.006, 0.015]). For the multi-speaker condition, direct post hoc comparison between target and distractor speech revealed that speech envelope tracking ($\beta = -0.007$, 94%HDI = [−0.013, −0.001]) was weaker for the distractor speaker compared to the target speaker. The same comparison for acoustic onset tracking points towards a similar effect ($\beta = −0.012$, 94%HDI = [−0.020, −0.005]). Overall, the results show stronger ocular speech tracking for attended speech compared to ignored speech in a single speaker and in a multi-speaker context (see Fig. 4a). They further indicate that in a multi-speaker context, eye movements track a target speaker more strongly compared to a distractor speaker. A summary of the statistics can be found in Supplementary Tables, Table 1.

TRFs revealed similar activation patterns for horizontal and vertical eye movements in Conditions 2 & 3 where the auditory modality is attended (see Fig. 4b). For the envelope, we observed positive initial peaks already at ~ 0 lag, with a fairly rapid decrease after ~ 200 ms back. For acoustic onsets, we observed a slower increase to a broader positive peak at ~ 200 ms with a slower decrease compared to the envelope TRFs. In general, TRF weights show a temporally more pronounced pattern for the speech envelope. We did not observe any meaningful weights to speech features when they were ignored in Condition 1. It is important to note that in this case TRFs can be interpreted not only based on their temporal profile but also on their directionality. Eye tracking data delivers positive and negative values on the horizontal and vertical plane (horizontal: + right, − left; vertical: + up, − down) that were preserved by the analysis. Thus, positive TRF peaks can be interpreted as shifts of gaze in the respective direction. However, it is important to note that TRF weights should be interpreted with great care. For one, our models also included control features that could potentially bias the

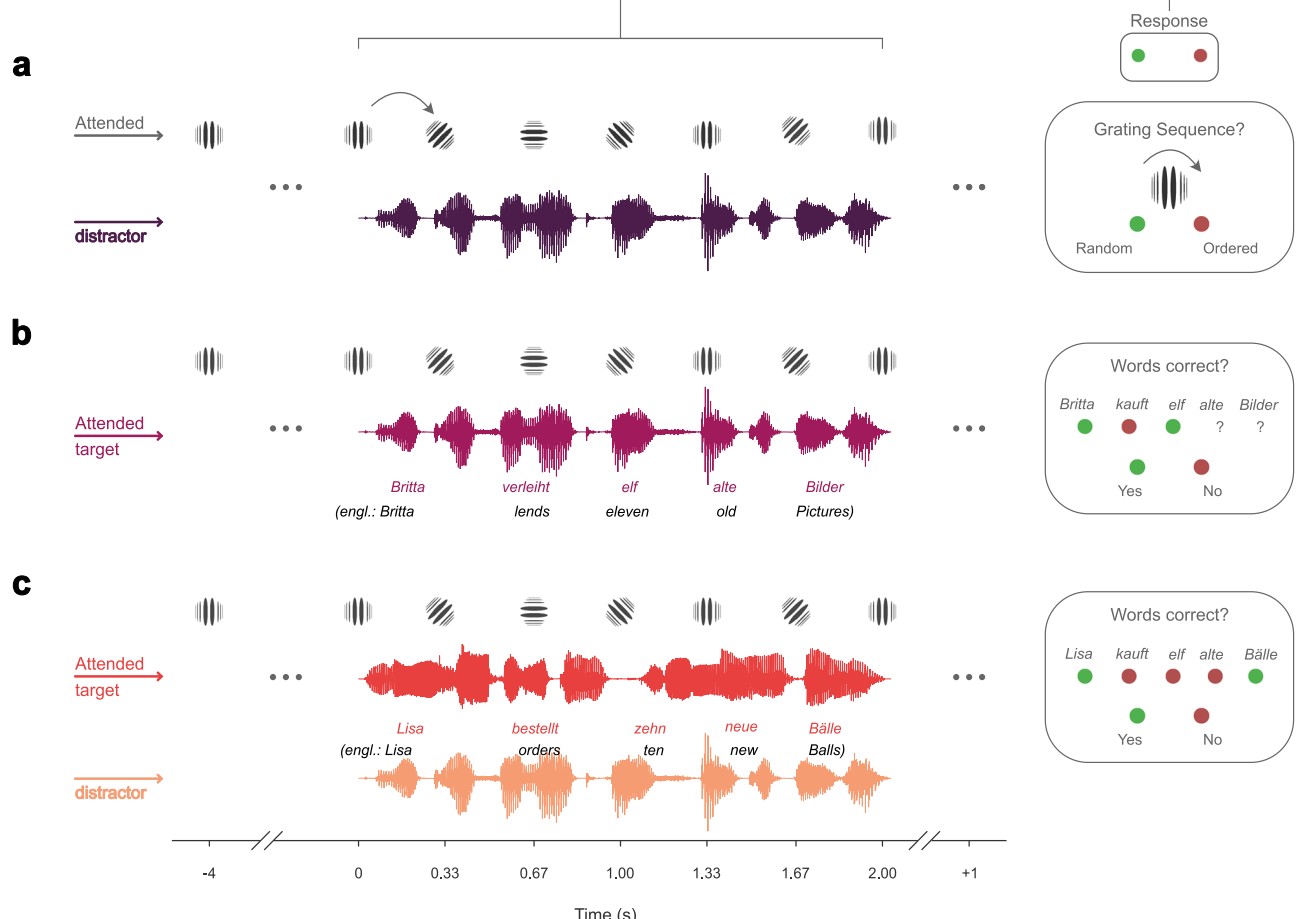

**Fig. 1 | The single-trial design for modulating selective attention to speech.** The task contained trials of short 5-word sentences of natural speech. Participants' attention was modulated within and across modalities. **a** In Condition 1, a rotating Gabor patch was attended in the visual modality, while speech served as a distractor (dark purple). **b** This was reversed in Condition 2, where speech was the focus of attention (bright purple). **c** In Condition 3, another speaker of the opposite sex was added to investigate the ocular speech tracking of a target speaker (red) with a simultaneously presented distractor (yellow). After each trial, participants responded to questions on the screen with a handheld button box (left button = green, right button = red): to the Gabor rotations in Condition 1 and to the presented words in the target speaker in Conditions 2 and 3.

spatial interpretability of TRFs. For another, the simplistic single-trial 5-word structure of the task could bias the temporal interpretability. In both cases, future studies with continuous unisensory designs are required for replication and validation.

## Ocular speech tracking is related to intelligibility

In response to a second question, we addressed the behavioral relevance of ocular speech tracking in terms of intelligibility and subjectively perceived listening effort. As intelligibility was probed for attended speech, i.e. where it was the target in a single speaker and a multi-speaker condition (see Fig. 1b, c), only these two conditions (multi vs. single speaker) were included in the Bayesian multilevel model. Fisher z-transformed prediction accuracies (z') of ocular speech tracking was included as the independent variable. We found a positive effect for the encoding of the speech envelope ($\beta = 19.113$, 94% HDI = [8.859, 29.223]) and acoustic onsets ($\beta = 11.695$, 94%HDI = [0.5705, 23.110]) on intelligibility indicating that, in the single speaker condition, higher intelligibility is reflected in stronger ocular speech envelope and acoustic onset tracking (see Fig. 5a). While we found a negative interaction with the condition of a multi vs. single speaker ($\beta = -13.224$, 94%HDI = [−25.212, -0.586]) for ocular speech envelope tracking, there was no substantial evidence for an interaction effect when using acoustic onset tracking as independent variable ($\beta = -5.380$, 94%HDI = [−17.822, 6.210]). This indicates that the link

between intelligibility and envelope tracking is decreased in a multi speaker condition (see Fig. 5b). There was no compelling evidence for an effect of neither ocular speech envelope ($\beta = -8.048$, 94%HDI = [−19.252, 3.663]) nor acoustic onset tracking ($\beta = 4.227$, 94%HDI = [−15.643, 7.834]) on subjectively perceived effort. A summary of the statistics can be found in Supplementary Tables, Table 2 and 3.

## Eye movements and neural activity share contributions to speech tracking

Thirdly, we asked whether eye movements and auditory selective attention share neural processes. As was shown by Popov et al.[16], a partial contribution of goal-driven oculomotor activity to typical cognitive effects in spatial auditory attention was retained even after removing scalp signal variance (e.g. by means of independent component analysis, ICA) related to ocular muscle activity. Based on their findings, it is important to address this possible contribution when evaluating neural responses in selective attention to speech, especially since our design did not entail any spatial discriminability or cues. With a mediation analysis approach (see Fig. 6a), we evaluated the relationship between eye movements and neural speech tracking with a cluster-based permutation test, contrasting the plain (c) against the direct (c') effects, i.e. model weights of the speech envelope for predicting neural responses on the one hand with model weights from an encoding model also including eye movements as an additional

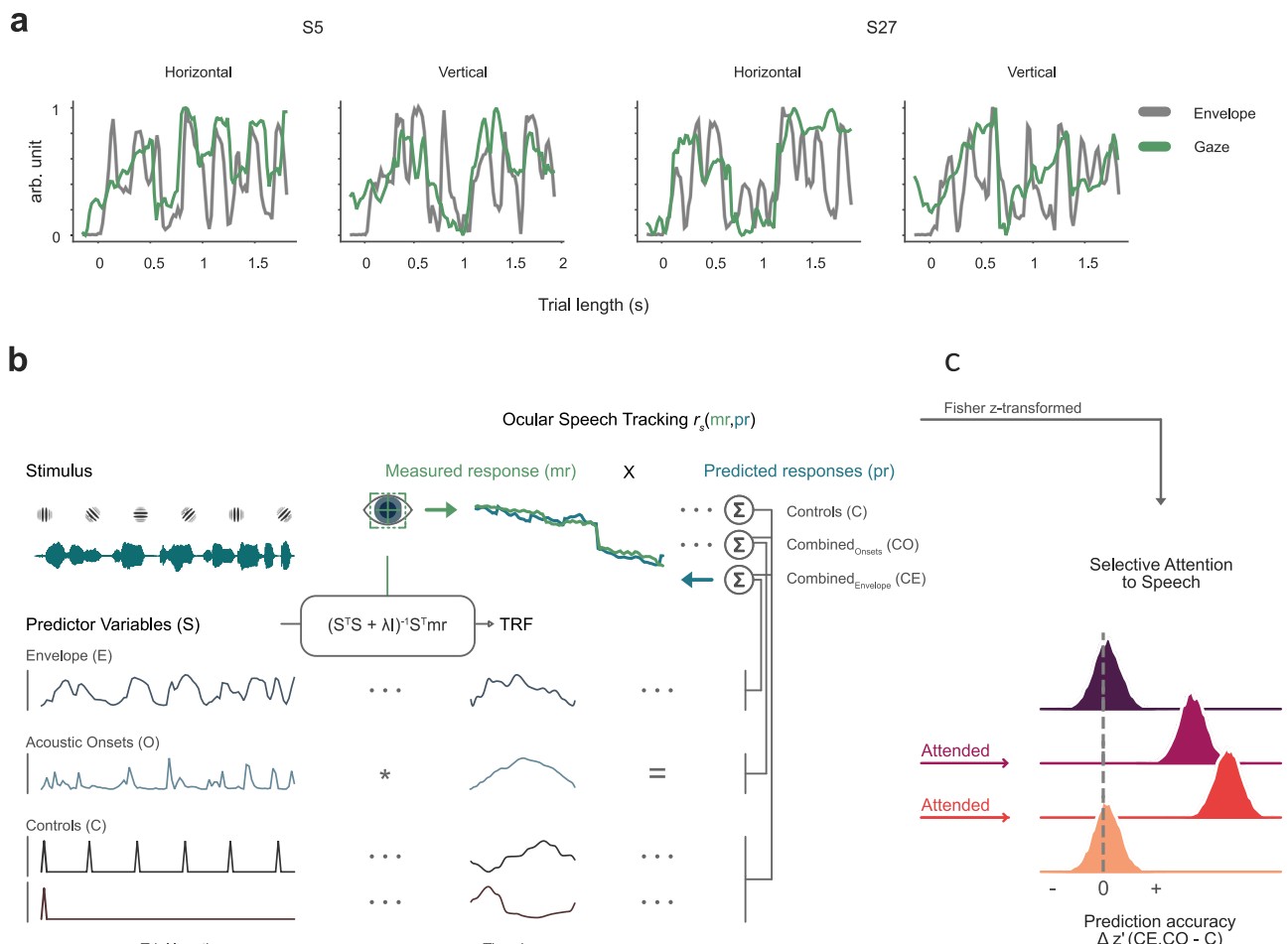

**Fig. 2 | The approach to establish the effects of ocular speech tracking.**
**a** Example trials of two participants (S5, S27) show how their measured gaze (green) on the horizontal and vertical plane follows envelope (grey) fluctuations (data was rescaled between 0 - 1 for illustration). **b** A regularized linear regression approach called temporal response functions (TRF) was used to predict how features of speech are tracked by eye movements. The difference in prediction accuracy for a control model (C) and combined models that additionally contained the speech envelope (CE) or acoustic onsets (CO) was used to estimate ocular speech tracking solely related to the acoustic features of interest, i.e. speech envelope and acoustic onsets. Prediction accuracies were calculated by Fisher z-transformed Spearman's rank correlations (z') between measured eye movements (mr) and predicted eye movements (pr). **c** We expected ocular speech tracking to be modulated by task-induced selective attention. The tracking difference between combined and control models (i.e. pure speech tracking) was expected to be higher whenever sentences were the target in a single-speaker or multi-speaker condition. For statistical computations, we used Bayesian multilevel regression models and illustrated the posterior distributions.

predictor on the other hand (see Fig. 6c, also see Mediation Analysis and Statistical Analysis). The tests revealed a significant difference (c' < c) for all three conditions with a left temporoparietal topographic pattern, partially overlapping with auditory processing areas. Eye movements shared contributions to neural speech processing mostly over left parietal sensors for the target in a single speaker condition ($t(29) = -4.40$, $p < 0.001$, Cohen's $d = -0.80$), the target in a multi-speaker condition ($t(29) = -4.90$, $p < 0.001$, Cohen's $d = -0.90$) as well as the distractor in a multi-speaker condition ($t(29) = -4.68$, $p < 0.001$, Cohen's $d = -0.85$, see Fig. 6c). In order to ensure that these effects did not merely stem from feature scaling or the inclusion of an additional predictor into the encoding model, we also compared the direct (c') effect against a model that included a shuffled version of eye movement data (across time, within condition). This led to almost identical results and therefore, crucially, the same conclusions (see Supplementary Figs., Fig. 3c).

## Discussion
Previous research established fundamental evidence for a joint network of attention and eye movements. In the auditory domain, several studies point towards interactions of the oculomotor system and

selective processing. The generalizability and validity of such interactions in complex, naturalistic environments have, to date, not been quantified. Here, we aimed to establish a direct link between ocular movements, selective attention to speech, and adaptive behavior. We further investigated the contribution of this ocular speech tracking to underlying neural processes. Using the sampled signal of continuous horizontal and vertical gaze activity in combination with TRFs, we show that eye movements track prioritized auditory features (i.e. envelope and acoustic onsets) in selective attention to speech. Crucially, ocular speech tracking differentiates between a simultaneously presented target and distractor speaker in the absence of any spatial discriminability and is further related to intelligibility. Moreover, using simultaneously recorded MEG data, we demonstrate that ocular speech envelope tracking contributes to the neural tracking effects of speech over left temporoparietal sensors, suggestive of auditory processing regions. Our findings provide insights into the encoding of speech in a joint network of auditory selective attention and eye movement control, as well as their contribution to the neural representations of speech perception. The description of this phenomenon raises several questions with regard to its underlying mechanisms as well as its functional relevance. They will need to be pursued in further

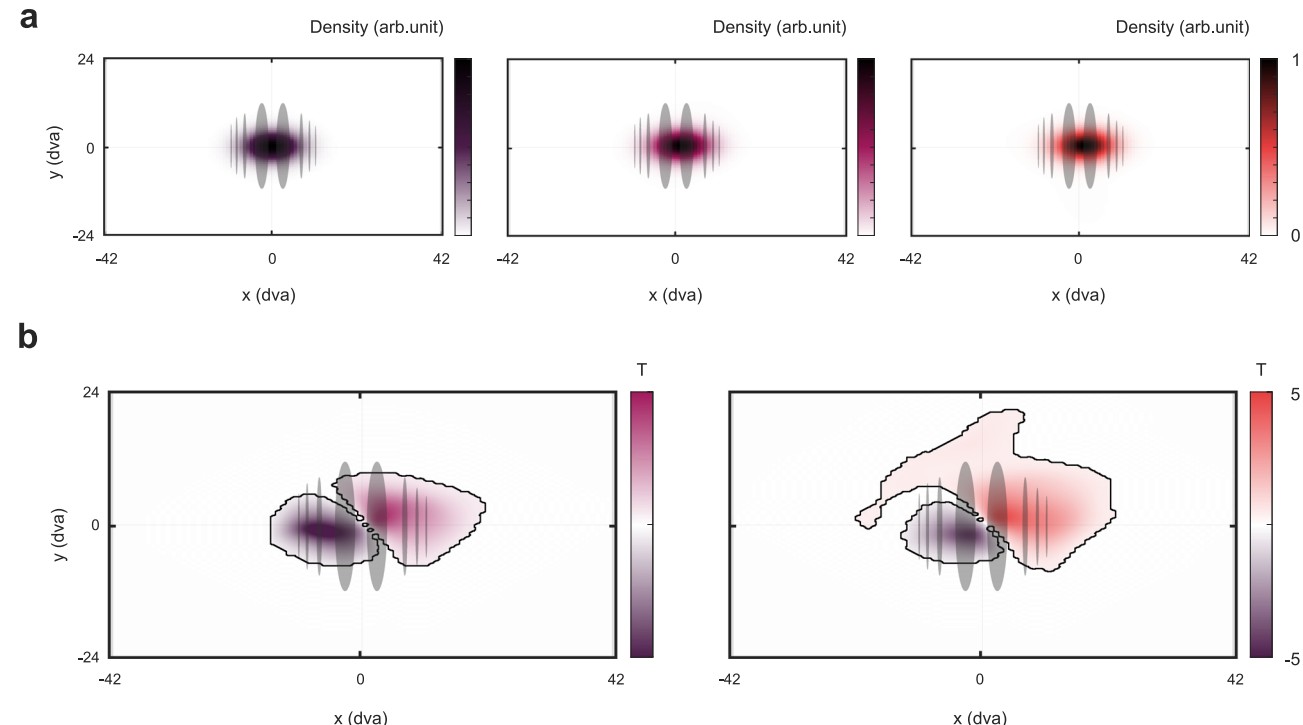

**Fig. 3 | Gaze behavior during sound presentations and Gabor rotations. a** An Average gaze density for the three conditions (from left to right: Condition 1 (dark purple), Condition 2 (bright purple), Condition 3 (red)). Screens are illustrated in degrees of visual angle (DVA). Density distributions validate that participants focused their gaze on the visual stimulus throughout all conditions of selective attention. **b** Cluster-based permutation tests (two-sided) on gaze density distributions show a slight shift of gaze to the top-right whenever the auditory modality is attended (Contrast Condition 2 vs 1: positive cluster $t(29) = 7.72$, $p < 0.001$, Cohen's $d = 1.41$, negative cluster $t(29) = -11.34$, $p < 0.001$, Cohen's $d = -2.07$. Contrast Condition 3 vs 1: positive cluster $t(29) = 6.26$, $p < 0.001$, Cohen's $d = 1.14$, negative cluster $t(29) = -8.69$, $p < 0.001$, Cohen's $d = -1.59$). Note that the 'widespread' cluster distributions result from low-density values that are not visible in a). $N = 30$.

studies. Nevertheless, we want to elaborate on a few possible principles that could be linked to ocular speech tracking.

In complex, naturalistic environments, gaze could aid the auditory system in prioritizing spectrotemporal acoustic information that reaches the ears. We argue that ocular speech tracking potentially reflects a learned coupling between the ocular and auditory systems to leverage precision in audiovisual perception. Changes in spectro-temporal acoustic information are inferred based on object identity and location and vice versa, where, in reference to an observer, the same object causes different sounds at different locations and different objects cause different sounds at the same location. In order to enhance perception in noisy environments, attended objects could inform the auditory system on spectrotemporal idiosyncrasies of the same leading to adapted neural firing along the auditory hierarchy while specific tonotopic activation alongside computations of inter-aural time and level differences could inform the visual system on redirections of gaze towards a target based on location and identity. Recent evidence in humans suggests that eye movements contribute to the computation of sound locations in relation to the visual scene at the very first stages of sound processing[11,12]. Similar studies with monkeys and cats suggest a midbrain hub of the inferior and superior colliculus (IC, SC) that affects auditory processing based on eye positions[8–10]. Barn owls engage the IC to create auditory space maps based on frequency maps and interaural time and level differences, integrating visual maps with cohesive sensory space maps in the optic tectum (the avian homologue of the SC[24,25]) under top-down gaze control[26]. The observed effect of ocular speech tracking could reflect the evaluation of attended sound based on spectral features and timing leading to the redirections of gaze that we report as ocular speech tracking. Based on TRFs, we see a systematic tracking of speech with a spatial shift of gaze to a top-rightwards direction within the boundaries

of the visual stimulus. Future studies are needed to further investigate whether this systematic shift is present also in other designs, especially with continuous speech designs (audiobooks). It would be intriguing to pursue the spatial dynamics of ocular speech tracking with spatially distributed or even spatially moving speakers. Here, one could also explore saccadic modulations of speech tracking (also on a neural level, see ref. 27) and further detail different eye movements (e.g. (micro)saccades and (micro)saccadic inhibition, slow-drift, smooth pursuit, etc.) and their effects on auditory prioritization and processing. However, with regards to the current design it should be noted that we used a non-spatial task without any meaningful visual stimulation where speech was always presented at the phantom center for both single and multi-speaker conditions. Ocular speech tracking could therefore be interpreted as a 'residual' activity based on learned associations between the spatiotemporal acoustics of speech and respective speakers.

It has been shown that barn owls recalibrate sound localization based on vision during their development[28], suggesting a learned alignment of auditory and visual stimuli based on a common source. Natural gaze behavior under the control of auditory attention could thus play an important role across species to navigate and interact with the environment, filtering and matching events or objects based on shared audiovisual spectrotemporal information. In humans, gaze activity could align to the acoustic features of attended speech to infer information about speaker identity and location (e.g. azimuth and distance) and match it with visual input. Speech, or verbal communication in general, has gained a central role as an advantageous survival strategy for social groups. Humans could exploit the ocular system during development to associate certain sound patterns with certain speakers, associating lip movements with sound and meaning, and ultimately guiding the development of speech in infants. Possibly,

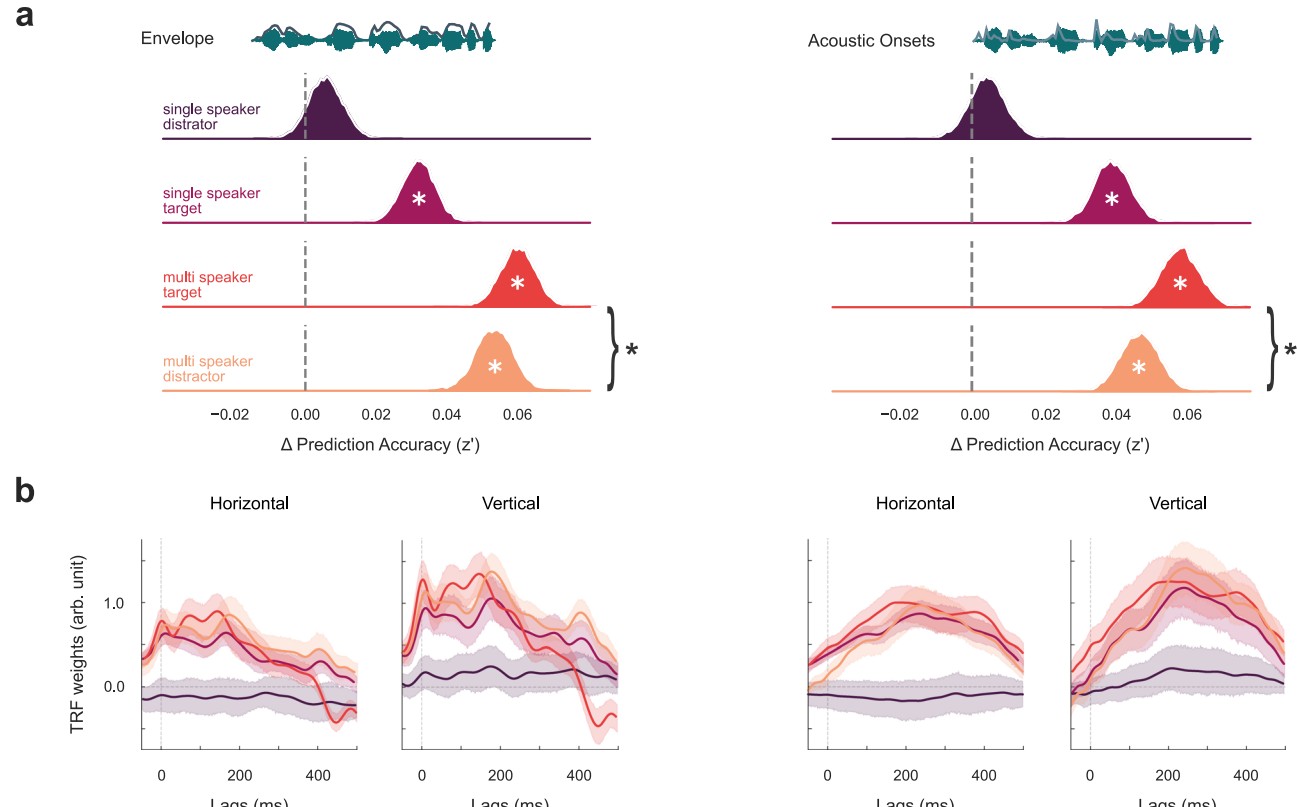

**Fig. 4 | The effect of selective attention on ocular speech tracking. a** Differences in Fisher z-transformed prediction accuracies (Δ z') between models that additionally included the speech envelope or acoustic onsets and a control model indicate significant tracking whenever speech was attended in a single speaker (bright purple) and multi-speaker (red) context. Post hoc comparison revealed stronger tracking of a target speaker compared to a simultaneously presented distractor (yellow) in the multi-speaker condition. No evidence for ocular speech tracking was found when speech was presented as a distractor to the visual modality (dark purple) **b** The temporal response functions (TRF) for speech envelope and acoustic onsets tracking. TRFs were resampled to 500 Hz for visualization. Center lines represent the mean, shaded areas represent 95% confidence intervals. Statistics were performed using Bayesian regression models. A '*' within posterior distributions depicts a significant difference from zero (i.e. the 94%HDI does not include zero). Curly brackets indicate post hoc comparisons: '*' = significant. N = 30.

selective attention and gaze support the prioritization of relevant acoustic information already at the cochlea via learned association of specific spectrotemporal activation patterns (for top-down modulation of the auditory periphery, see refs. [29–31]). This could aid the differentiation between speakers (e.g. a female compared to a male voice) and support the temporal alignment observed in stronger ocular speech tracking for a target in the multi-speaker condition even in absence of visual or spatial discriminability. Further studies could investigate potential effects/benefits of matching visual input (videos), e.g. lip movements, on the phenomenon.

The idea of an active sampling strategy of spatiotemporal information is further supported by the temporal dynamics of ocular speech tracking. TRFs show a first, initial peak around zero lag (see Fig. 4b), potentially indicative of a supportive mechanism of ocular speech tracking at the very first stages of sound processing to aid prioritization of overlapping spatiotemporal information. This would also align with the results on the shared contribution of ocular and neural activity to speech tracking at left temporoparietal sensors, indicative of auditory processing areas (see Fig. 6c). Such an immediate engagement of the ocular system would further suggest a complementary predictive processing account. Anticipation and accurate allocation of events in time have been found for language processing[32] and eye movements in motion perception[33–36]. Predictive mechanisms should lead to a reduction in processing costs, thus interindividual differences in anticipatory TRF peaks could be related to subjectively perceived listening effort. However, we would like to point out that the presented study design limited the analysis and interpretation of TRFs due to its short 5-word sentence structure. For one, anticipatory effects could be biased by the highly predictable syntactic structure of the sentences. This assumption is further supported by neural TRFs (see Fig. 6b), also showing relatively early (pre)activation patterns. Secondly, sentences were too short to allow for wider TRF windows that could give more detailed information about later dynamics > 500 ms where a clear differentiation of target and distractor in a dual speaker mixture seems to take place. Future studies should investigate the precise temporal dynamics of ocular speech tracking and potential predictive processes in continuous designs (e.g. with audiobooks).

Ocular speech tracking could also relate to a more general engagement of the motor system in support of speech perception. Recent findings suggest a link between rates of eye movements during text reading and typical speech production/perception rates[37]. We observed a general right lateralized bias of gaze whenever the auditory modality (i.e. speech) was attended (see Fig. 3b). TRFs point towards a successive top-rightwards shift of gaze with similar timing aspects compared to a reading of ~ 200 ms. Since ocular speech tracking effects suggest a temporal alignment of this right lateralization with speech features, it could be argued that our eyes move with the speech streams as if the words were read as text. If this was the case, we would expect a shift of gaze towards the left side for cultures that read text from right to left. This would render ocular tracking specific to (1) humans, (2) speech, and (3) cultural context. Future studies should further explore the spectrotemporal characteristics with regards to reading by applying similar analyses to designs (1) with animals (that also use verbal communication, e.g. birds), (2) tone sequences, (3) across cultures.

**a**

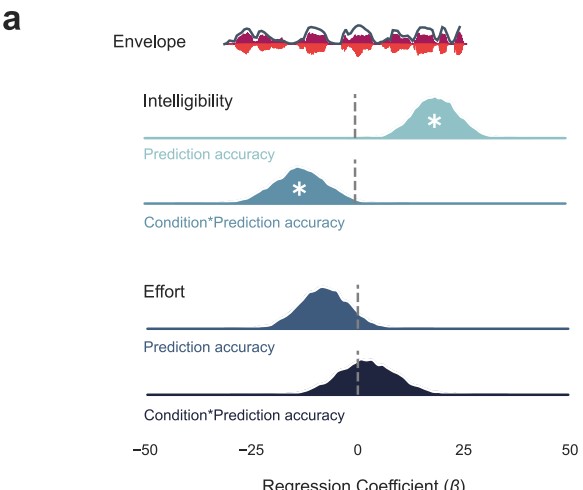
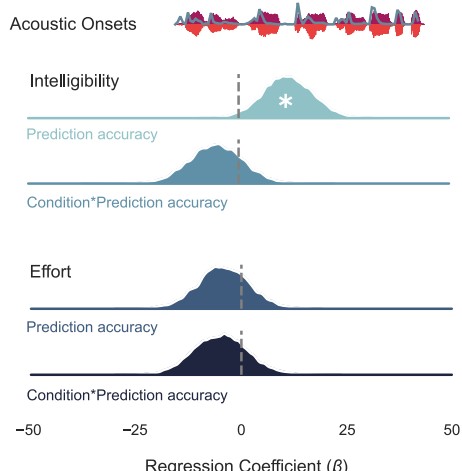

**b**

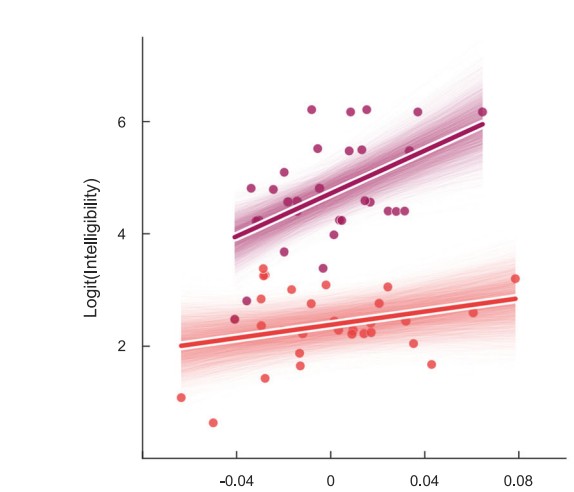
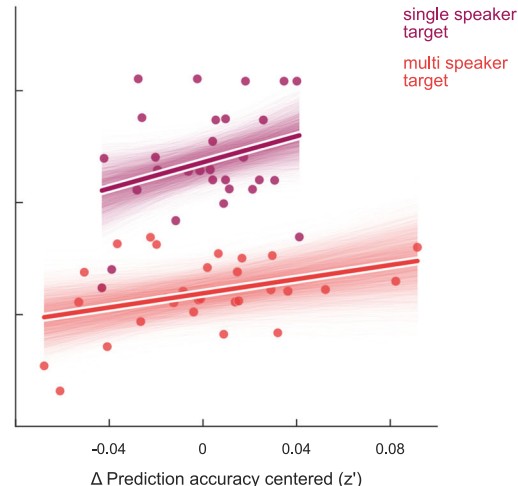

**Fig. 5 | Ocular speech tracking and its relation to speech intelligibility and subjective listening effort.** Intelligibility was probed only for attended speech. We therefore used intelligibility and subjective listening effort scores for targeted speech in the single (dark purple) and multi-speaker (red) context to assess its relation to ocular speech tracking. **a** Ocular speech tracking (differences in Fisher z-transformed prediction accuracies (Δ z')) was related to intelligibility with an interaction effect only for the speech envelope. There was no substantial evidence for a relation to subjective effort. **b** Both ocular speech envelope and acoustic onset tracking predict intelligibility (logit-transformed). The interaction effect for ocular speech envelope tracking indicates that the link between intelligibility and envelope tracking is decreased in a multi-speaker condition, while for acoustic onsets the effect seems to be similar in both conditions (center lines represent the mean, shaded areas represent the 94%HDI, dots represent participants). Statistics were performed using Bayesian regression models. A '*' within posterior distributions depicts a significant difference from zero (i.e. the 94%HDI does not include zero). *N* = 30.

Another explanation for the observed ocular speech tracking effects could be a general push-pull process of task dis-/engagement, i.e. gaze aversion of the visual modality to free up resources and facilitate the processing of auditory information[38–41]. Thus, whenever the task is to listen closely, or if listening becomes increasingly difficult, we move our eyes away to attenuate interfering visual input (note that complete eye closure seems to increase alpha modulations by auditory attention, but does not, however, improve listening behavior[42]. Also, internal attention in insight problem-solving seems to relate to increased blinking and off-center gaze shifts[43]). Gaze densities during stimulation periods (see Fig. 3a) show a slight shift off-center as well as higher variance in conditions where participants had to attend the auditory modality (see Fig. 3b). This supports the assumption of a more general disengagement from the visual modality for auditory processing, arguably to free up processing resources. However, participants also could shift their gaze slightly away from the distracting visual stimulus to free up resources for a more precise evaluation of temporal speech features. This could support the processing of overlapping spatiotemporal features, especially in the multi-speaker condition, since the visual stimulus was meaningless for auditory processing. TRFs show positive weights peaking at ~ 200 ms without any pronounced negative weights within the analyzed time window. This indicates a successive shift of gaze in a top-right directionality. In addition, we verified that participants kept their gaze within the boundaries of the distracting visual stimulus throughout the stimulation period (see Fig. 3a). Altogether, the spatial dynamics of ocular speech tracking seem to be too timing-specific for a general gaze aversion process. Future studies could directly address the potential principle of gaze aversion by implementing an eyes-closed condition. It should be noted that EOG activity seems to align with attended acoustics also in an eyes-closed condition[19]. In general, the post hoc lateralization findings of ocular speech tracking during a non-spatial task without any meaningful visual stimulation require further replication and validation. As of now, the mechanistic explanation of a

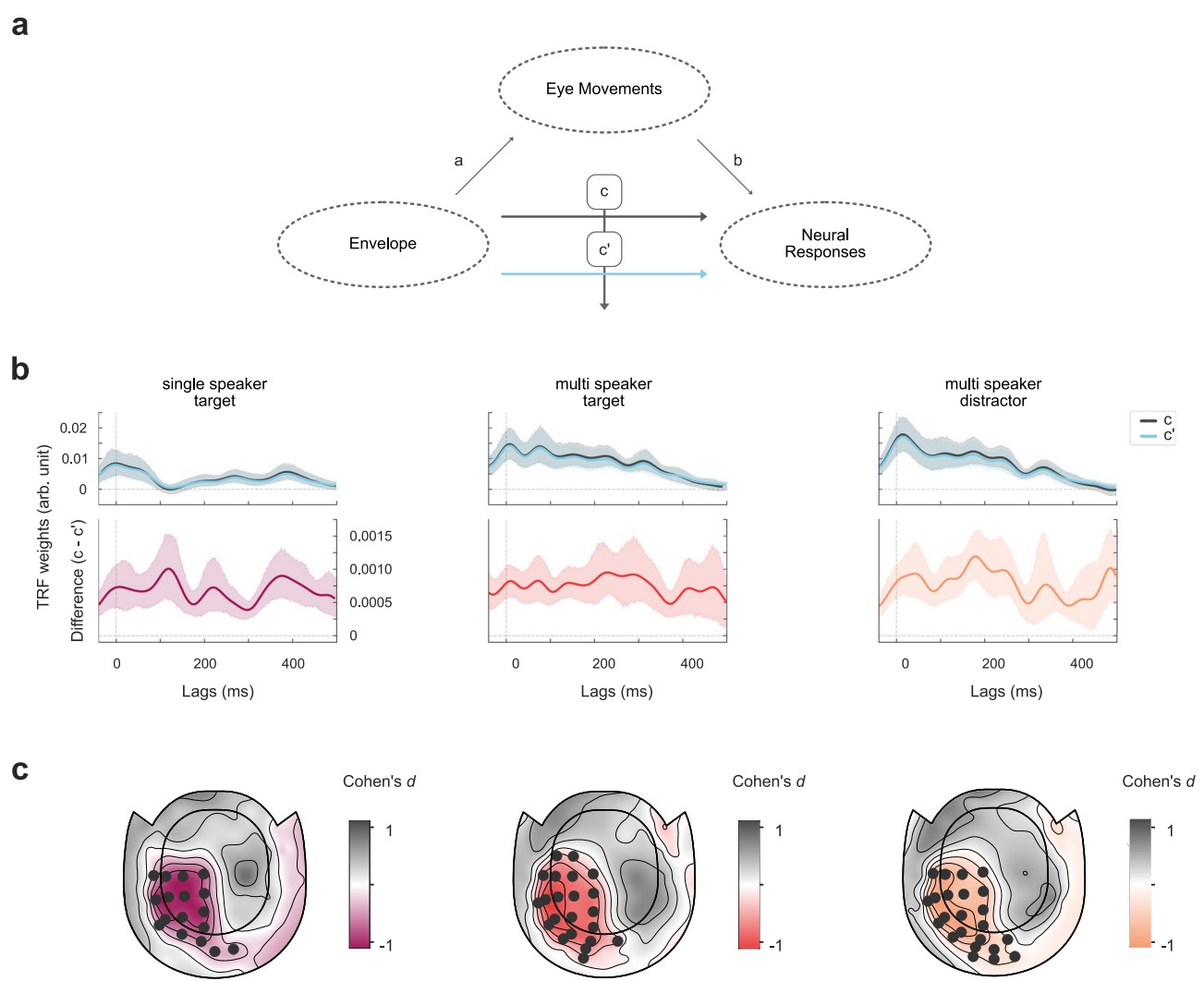

**Fig. 6 | The approach to establish shared contributions of eye movements and neural activity to speech tracking. a** With a mediation analysis approach, we investigated the shared contribution of eye movements and neural responses to the speech envelope encoding. For this, we compared the plain effect (c) of the speech envelope on neural activity to its direct (residual) effect (c') by including an indirect effect via eye movements. The two models described above were calculated to predict neural responses of all 102 magnetometer channels separately. **b** Exemplary temporal response functions (TRF) for the plain effect (c) and direct (residual) effect (c') and respective differences (c – c') to visualize the mediation effect over time. For illustration, we chose the channel that showed the highest prediction accuracy for the plain effect (c) in the single-speaker target condition ($r_s = 0.12$). TRFs were resampled to 500 Hz for visualization. Center lines represent the mean, shaded areas represent 95% confidence intervals **c** We used a cluster-based permutation-dependent *t*-test to compare TRFs (using absolute values) of both models at each sensor and report effect size Cohen's *d* averaged over sensors within significant clusters. Cluster-based permutation tests (one-sided) for the contrasts (c' < c) revealed a small mediation effect by eye movements for the relationship between the speech envelope and neural responses over left parietal sensors in the single speaker condition (bright purple; $t(29) = -4.40$, $p < 0.001$, Cohen's $d = -0.80$), in the multi-speaker condition for a target ($t(29) = -4.90$, $p < 0.001$, Cohen's $d = -0.90$) as well as a distractor speaker(yellow; $t(29) = -4.68$, $p < 0.001$, Cohen's $d = -0.85$). Marked sensors in topographies belong to sensor clusters on the basis of which the null hypothesis of no difference was rejected. $N = 30$.

residual activity based on learned association seems to be the most applicable to the findings in the current study.

Taken together, we propose several potential principles of ocular speech tracking that need to be evaluated in greater detail by future research. Continuous speech designs (e.g. audiobooks) could be utilized to replicate the present findings and further investigate the precise temporal dynamics of the reported effects. In turn, potential interactions with predictive processes as well as interactions with behavioral markers like effort and intelligibility could be quantified. Neurophysiological evidence in animals for similar interactions of eye movements and auditory processing further urge the question of whether ocular speech tracking displays a learned association in beings with complex verbal communication structures like humans, or whether it represents a general ocular tracking of the acoustic environment as adaptive behavior across species. It will thus be important

for future research to identify the underlying mechanisms of the observed effects. Potential links to alpha modulations[16,44–46] or arousal states with regards to (sub)cortical neurotransmitter-dependent modulations[47] could be investigated. Along these lines, it will be important to investigate how other measures of ocular activity could be predicted by the speech envelope (and other features), including blinks, pupil dilation, and (micro)saccades (also see ref. 19). However, many trials did not include the mentioned responses of interest (blinks and (micro)saccades), which makes any analysis in this direction problematic (along with its interpretation). Additionally, the rotating Gabor sequence on the screen during the stimulation period severely influenced the pupil response. Any dilation measure would thus be so heavily confounded by the visual stimulation that potential responses to speech could not be interpreted with great confidence. We therefore strongly encourage replicating this study in a continuous design

without any visual stimulation. Future studies should take advantage of the findings presented here and define a region of practical equivalence (ROPE) to further infer support for any potential null-hypothesis testing[48].

In response to a second question, we addressed ocular speech tracking and its relation to adaptive behavior. We did not find a relationship between ocular speech tracking and subjectively perceived effort, which questions the previous assumption of gaze aversion for the reallocation of processing resources. Either effort is not reflected in the engagement of the ocular system in speech tracking, or the measure of effort was not sensitive enough (alternatively, neural measures of effort like alpha modulations[49,50] or pupillometry could be used to estimate effort in more detail, for dynamic modeling of pupil data also see ref. 51). Also, no difference in general task engagement was found (see Supplementary Methods). Instead, we found that ocular speech tracking is related to intelligibility. Our results are supported by findings on increased intelligibility for a spatially discriminable target speaker in a multi-speaker mixture when the eyes point towards the target location[17]. It therefore seems likely that eye movements and intelligibility of speech are, in fact, related. Taking the possible interpretations for the ocular speech tracking effects into consideration, (1) improved prioritization of relevant spectrotemporal information could improve the intelligibility of a target speaker in the multi-speaker condition, and (2) further supported by predictive processes. (3) An engagement of the motor system could support phonological transformation processes to increase speech perception, and (4) disengagement from the visual modality, i.e. gaze aversion could free resources for speech processing to improve intelligibility. Further research on the topic is needed to get a better understanding of the interaction between ocular speech tracking and intelligibility, also on a neural level. In addition, we cannot entirely rule out a potential contribution of individual differences in attention and working memory to the behavioral measure of intelligibility. Future studies should address this possibility in greater detail.

As a final step, we investigated the potential contribution of eye movements to neural differences typically observed in selective attention to speech. Our assumption that ocular speech tracking and selective attention to speech share underlying neural computations was based on recent findings by Popov et al.[16] who demonstrated a partial contribution of goal-driven oculomotor activity to typical cognitive effects in spatial auditory attention. It was therefore important to address a possible contribution in the present work since we previously established the effects of ocular speech tracking. Using a mediation analysis approach (see TRF Model Estimation and Data Analysis), we provide evidence for a shared contribution of eye movements and neural activity to speech processing over left temporoparietal areas in a single and multi-speaker context. We thus observe a general contribution to sensors indicative of auditory processing regions, which gains importance when considering recent evidence on the ocular modulation of neural excitability of cortical auditory regions[27]. To this end, we would like to specify that this exploratory analysis needs to be confirmed by future studies in a more detailed and methodologically tailored manner. Also, it should be noted that the current findings do not imply a clear directionality or causality - it is possible that the neural activity in question is used to support sound-related ocular activity or vice versa. Following studies that solely use continuous speech could focus on relating eye movements and alpha oscillations in selective attention to speech and establishing concrete evidence on the temporal dynamics of this interaction on a source level. As we used a multisensory single-trial design, we believe a continuous unisensory approach would be more suitable for this kind of analysis. Here, we provide a first step in this direction, highlighting a shared contribution on a sensor level that needs to be taken into consideration in future research on auditory cognition.

In summary, the present report establishes a hitherto unknown phenomenon, ocular speech tracking, which enables the monitoring of prioritized auditory features in selective attention to natural speech. Crucially, ocular speech tracking is stronger for a target compared to a distractor in a multi-speaker condition and is related to intelligibility. Moreover, our results suggest a shared contribution of oculomotor and neural activity to speech processing that needs to be taken into consideration in future research on auditory cognition. The present work extends previous findings of a joint network of attention and eye movement control and offers research directions towards the neurobiological mechanisms of the phenomenon, its dependence on learning and plasticity, as well as its functional implications in social communication.

## Methods

### Participants

30 healthy participants (19 female, $M_{age} = 26.27$, $SD_{age} = 9.08$) were recruited for this study. Participants were compensated either financially or via course credits. We set the sample size based on previous publications from our lab with similar setups and designs (e.g. refs. 52,53). All participants were German native speakers and reported normal hearing, and (corrected to) normal vision. Participants gave written self-reports on their sex and gender along with informed consent and reported no previous neurological or psychiatric disorders. The experimental procedure was approved by the ethics committee of the University of Salzburg and was carried out in accordance with the Declaration of Helsinki.

### Experimental procedure

The experiment lasted ~ 3.5 hours. Five head position indicator (HPI) coils were first applied to the participants' scalp. Anatomical landmarks (nasion and left/right pre-auricular points), HPI locations, and around 300 additional head shape points were then sampled using a Polhemus FASTTRAK. Recording sessions started with 5 min of resting state data (2.5 min eyes open / closed), followed by two blocks of passive listening to tone sequences of varying entropy levels (as in ref. 52). Afterwards, one block of 10,000 clicks at 60 dB sound pressure level was presented to determine individual auditory brainstem responses (as in ref. 53) while participants watched a landscape movie (LoungeV Films, 2017). As these parts of the experiment relate to separate research questions, they are not explained in further detail here. The main task (Fig. 1) consisted of three conditions split into six blocks of 50 trials, i.e. 100 trials per condition. The order of the blocks and trials was randomized across participants. The purpose of the three conditions was to modulate attention within as well as across modalities. Condition 1 (see Fig. 1a) tasked the participants with attending to the visual modality (regularity of Gabor rotations) while being distracted by a short sentence spoken by a male voice. Condition 2 (see Fig. 1b) reversed the task, requiring participants to allocate attention to the auditory modality (natural spoken sentence) while visual stimulation served as a distractor. In Condition 3 (see Fig. 1c), the visual modality and the male speaker distracted the participants from attending to an added female target speaker. Each trial started with a silent 4 s prestimulus interval during which participants had to keep their gaze on a Gabor patch presented in the center of the screen. Then, a short sentence was played while simultaneously the Gabor patch on the screen tilted to one of four perceptually different angles (0°, 45°, 90°, 135°). For the duration of the sentence, the Gabor patch was tilted with a fixed stimulation rate of 3 Hz, each lasting for 100 ms. The tilting either followed a) an ordered, clockwise sequence where the upcoming Gabor was tilted 45° with a probability of 75% or stayed at the same angle at 25% probability, or b) was randomly tilted to one of the four predefined angles, all equally likely with 25% (note that the transitional probabilities and stimulation rates were the same as in the passive listening task and chosen to not interfere/co-occurre with

common syllable rates in language at ~ 4 Hz). Ordered and random sequences were pseudorandomized across trials. Stimulation offset was followed by a 1 s silent poststimulus interval with the Gabor patch at its original tilt at 0°. During the whole trial period, participants were instructed to keep their gaze on the Gabor patch in the center of the screen - regardless of condition - to allow for valid eye-tracking data (also see Fig. 3a). Each trial was followed by a behavioral response with a question on the stimulation period presented on the screen. In Condition 1, the response required participants to successfully infer whether the Gabor transitions followed an ordered or random sequence. In Conditions 2 and 3 we assessed intelligibility scores, probing participants on every word in the attended sentence. For this, we randomly replaced up to all five words of the sentence during the stimulation period (see Stimuli) and presented them on the screen. Participants could then mark every word as 'yes', i.e. correct, or 'no', i.e. false. Every word on the response screen could have potentially been correct or false. At the end of each block, we additionally assessed task engagement and subjectively perceived effort on a 5-point Likert scale. All responses were given on a handheld button box. All auditory stimuli were presented binaurally at the phantom center at a comfortable loudness level. The experiment was coded and conducted with Psychtoolbox-3[54,55] implemented in Matlab R2020b[56] with an additional class-based library ('Objective Psychophysics Toolbox', o_ptb[57]).

## Stimuli

The visual stimulation was a Gabor patch (spatial frequency: 0.01 cycles/pixel, sigma: 60 pixels, phase: 90°). For auditory stimulation, we used 100 sentences from the 'Oldenburger Satztest' (OLSA[58]) for the male speaker. We created 100 additional 'surrogate' sentences in the same style for the female speaker. Alongside randomization, this ensured that any effects, especially in the multispeaker Condition 3, could not be attributed to memorization of previous trials. Often used in studies on hearing impairment, the OLSA is a standardized audiometric test to assess speech intelligibility. It features lists of 5-word sentences in a fixed form: Name - Verb - Number - Adjective - Noun (for example "Peter verleiht vier kleine Bilder.", in engl.:"Peter lends four small pictures.", also see Fig. 1). Ten words of each word type are used to create 100 unique sentences through random combinations. For the 'surrogate' sentence list, we substituted the ten words per word type with ten other Names, Verbs, Numbers, Adjectives, and Nouns respectively (for example "Karin bestellt zehn blaue Körbe", in engl. "Karin orders ten blue baskets"). This led to 100 unique sentences for the male speaker (target in Condition 2, distractor in Conditions 1 & 3) and 100 unique sentences for the female speaker (target in Condition 3). Unlike commonly used questions on general content or last words in speech tracking designs with longer segments (e.g. audiobooks), the fixed 5-word structure allowed us to probe speech intelligibility on a word-by-word level. To synthesize the 200 extracted sentences into natural-sounding speech, we used the IBM Watson text-to-speech service (TextToSpeechV1 package). We synthesized German text-to-speech at a sampling rate of 44.1 kHz using the implemented voices for the male speaker (voice 'de-DE_DieterV3Voice') with adjusted prosody rate to −10 % in order to match the female speaker's ('de-DE_ErikaV3Voice') syllable rate. This led to slightly different sentence durations for the male and female speaker ($M_{male}$ = 2.02 s, $SD_{male}$ = 0.16, $M_{female}$ = 2.22 s, $SD_{female}$ = 0.13) due to the slightly longer surrogate sentences (as the number of words was exhausted from 1–10 in the original list, for surrogate sentences the number words included thirteen, fourteen,...). However, this was controlled for later in the analysis by cropping the aligned data (see TRF Model Estimation and Data Analysis) in all conditions to the respective shorter trials of the multispeaker Condition 3, resulting in equal durations for both speakers. In addition, rare hardware buffer issues during the experiment led to additional noise in the stimulation for some participants. We excluded those trials from later analysis and randomly subsampled the same

amount of trials for all other participants. In sum, 98 trials per condition were retained for further analysis.

## Data acquisition and preprocessing

MEG data were simultaneously acquired alongside ocular data at a sampling frequency of 10 kHz (hardware filters: 0.1–3300 Hz) with a whole head system (102 magnetometers and 204 orthogonally placed planar gradiometers at 102 different positions; Elekta Neuromag Triux, Elekta Oy, Finland) that was placed within a standard passive magnetically shielded room (AK3b, Vacuumschmelze, Germany). For further data processing, a signal space separation algorithm implemented in the Maxfilter program (version 2.2.15) provided by the MEG manufacturer was used to remove external noise and realign data from different blocks to a common standard head position. Afterwards, we preprocessed the data using Matlab and FieldTrip. At first, 10 kHz data were resampled to 1000 Hz for further computations using the default implementation in FieldTrip (cutoff frequency = 500, kaiser window FIR filter, order: 200). Then, a bandpass filter between 0.1–40 Hz was applied (zero-phase FIR filter, order: 16500, hamming window). To remove ocular (horizontal and vertical) and cardiac artifacts, 50 components were identified from each experimental block using runica-independent component analysis (ICA). Components originating from eye movements and heartbeat were then identified by visual inspection and removed. Artifact-free brain data was then cut into epochs from -1 to 4 s around stimulus (i.e. speech) onset and corrected for a 16 ms delay between trigger and stimulus onset generated by sound traveling through pneumatic headphones into the shielded MEG room. Eye-tracking data from both eyes were acquired at a sampling rate of 2 kHz using a Trackpixx3 binocular tracking system (Vpixx Technologies, Canada) with a 50 mm lens. Participants were seated in the MEG at a distance of 82 cm from the screen, with their chin resting on a chinrest to reduce head movements. Each experimental block started with a 13-point calibration and validation procedure that was then used throughout the block. Blinks and saccades were automatically detected by the Trackpixx3 system and excluded from horizontal and vertical eye movement data. Subsequently, data were preprocessed in Matlab R2020b. Position data from left and right eyes were averaged to increase the accuracy of gaze estimation[59]. We then converted data from pixel to visual angle in degrees. Gaps in the data due to blink and saccade removal were interpolated using a piecewise cubic Hermite interpolation. Artifact-free gaze data was then imported into the FieldTrip Toolbox[60], bandpass filtered between 0.1–40 Hz (zero-phase finite impulse response (FIR) filter, order: 33000, hamming window), resampled to 1000 Hz and cut into epochs from -1 to 4 s around stimulus (i.e. speech) onset. Finally, we corrected again for the 16 ms delay between trigger onset and actual stimulation.

To further calculate gaze density during trial periods, we followed the same analysis procedure as in ref. 16. In short, a 2D density histogram was created after multiplying each sample point of gaze on the horizontal and vertical plane with a Gaussian filter.

## Predictor variables for TRF models

**Controls.** We included control predictors for eye responses to visual (Gabor) onsets according to the fixed 3 Hz presentation rate throughout the sentence and pure auditory (speech) onsets by adding intercepts (i.e. impulse trains) at respective timings.

**Envelope.** Both auditory predictors (Envelope & Acoustic Onsets) were based on gamma tone spectrograms of the 200 natural-sounding speech sentences. Spectrograms were calculated over 256 frequencies, covering a range of 20–5000 Hz in equivalent rectangular bandwidth space[61], resampled to 1000 Hz, and scaled with exponent 0.6[62] using Eelbrain[63]. The 1-Band Envelope was then derived by taking the sum of gamma tone spectrograms across all frequency bands[63], thus reflecting the broadband acoustic signal.

**Acoustic onsets.** Additionally, we derived acoustic onsets by applying a neurally inspired auditory edge detection transformation to the gamma tone spectrogram[64] using the publicly available 'TRF-Tools' (https://github.com/christianbrodbeck/TRF-Tools) edge detection implementation for Python, with default settings and saturation scaling factor of c = 30. Again, 1-band Acoustic Onset representations were obtained by taking the sum across all frequency bands.

All predictors (see Fig. 2b) were resampled to 1000 Hz for subsequent alignment and analysis, to match with the sampling frequency of eye-tracking data.

## Model comparisons

In order to estimate ocular tracking of the speech envelope and acoustic onsets, we chose to include a control model (C) in the analysis that uses visual onsets and trial/speech onsets as predictors (see Fig. 2b), as they confounded the responses to the speech features of interest. Using the prediction accuracy of this control model as a basis, we then combined the control predictors with one of the speech features, leading to two combined models controlling for visual and trial onsets and entailing the speech envelope (CE) or acoustic onsets (CO) as predictors. In order to obtain the predictive power solely related to the speech features, we then subtracted the prediction accuracies of the control model from those of the combined models. This was done separately for every participant in each condition, resulting in a 'pure' prediction accuracy value Δ z' (based on Fisher z-transformed Spearman rank coefficients, see TRF Model Estimation and Data Analysis) as an estimate of ocular speech tracking (see Fig. 2c).

## TRF Model Estimation and Data Analysis

Prior to model computations, preprocessed eye-tracking and MEG data were temporally cut and aligned to the corresponding predictor variables (the blink and saccade rate, and therefore the number of samples that were interpolated for later analysis, was low: M = 5.00%, SD = 4.27% for blinks and M = 1.13%, SD = 2.81% for saccades respectively). Then, aligned trials were downsampled to 50 Hz for TRF model estimation after an antialiasing low-pass filter at 20 Hz was applied (zero-phase FIR filter, order: 662, hamming window). Impulse trains, i.e. control predictors, were then restored by adding "1 s" at the nearest time points of original sampling rate onsets without applying any filters to avoid artifacts. We chose to downsample to 50 Hz as the most relevant power modulations of speech and attention do not exceed 20 Hz (i.e. 2 ½ * the sampling rate).

To further probe ocular speech tracking under selective attention, we used a system identification technique called temporal response functions (TRF) as implemented in, and provided by, the open-source mTRF-Toolbox[22,23] for Matlab. In short, TRFs pose time-resolved model weights to describe a stimulus-response relationship (forward / encoding models), e.g. how features of speech are transformed into responses at multiple time-lags. Whereas this technique is usually used to model neural responses, here we exploited this approach and applied TRF models on eye tracking data to investigate the relationship between speech features and eye movements (see Fig. 2b). In the present study, we used ridge regression, a regularized linear regression approach, at a time-lag window of -100 to 550 ms to compute encoding models, following a leave-one-trial-out cross-validation procedure to control for overfitting. This means we used all but one trial (of the 98 per condition) to estimate TRFs for a set of stimulus features (i.e. predictors) that were in turn applied to those of the left-out test-trial to obtain a predicted ocular response. Prediction accuracy was then evaluated by calculating a Spearman's rank correlation between the originally measured, preprocessed response and the predicted response by the model (note that forward models make predictions independently for each channel, which here refers to horizontal and vertical eye movement 'channels' in eye-tracking data, as well as 102 magnetometers in neural data in the next section). Before model

estimation, predictors and responses were scaled by their respective ℓ1 norms. After each trial had been the test-trial once, prediction accuracies were Fisher z-transformed and then averaged over trials. This resulted in one Fisher z' value, for horizontal and vertical eye movements respectively, that describes how well the TRF model could predict an ocular response of a particular participant to speech in differing conditions of attention, which renders prediction accuracy a measure of ocular speech tracking (note that for statistical analysis we averaged prediction accuracies for horizontal and vertical eye movements once the difference between combined and control models was calculated, also see Model Comparison). The modeling procedure described above was carried out for every condition of selective attention to speech in the presented paradigm (see Fig. 1), i.e. when a single speaker was the distractor in Condition 1, a single speaker was the target in Condition 2, a speaker was the target in a dual speaker mixture in Condition 3, a speaker of opposite sex was the distractor in a dual speaker mixture in Condition 3.

To further control for overfitting, ridge regression includes a regularization parameter λ that penalizes large model weights (for a detailed explanation of the ridge parameter, see ref. 23). We empirically validated the optimal ridge parameter by using a nested cross-validation procedure (i.e. within every n-1 training set another leave-one-out cross-validation was used to obtain model results for different λ values). This procedure was carried out over a range of λ values of $10^{-5}$–$10^5$ (in steps of $10^1$) for each model (see Model Comparisons) over all participants and conditions. The final optimal lambda value for a certain model was then obtained by averaging the mean absolute error of the cross-validation over all trials, channels, conditions, and participants. We consequently chose the lambda value that led to the lowest mean absolute error. Based on this procedure, a single optimal lambda value of λ = $10^{-3}$ was used to estimate speech encoding in ocular activity for all encoding models (see Model Comparisons) for all conditions of selective attention.

Following this analysis of ocular speech tracking, we established the behavioral relevance of this effect. As relevant dependent variables for the statistical analysis, we calculated individual intelligibility and effort scores from participants' behavioral responses. Intelligibility scores were calculated as the sum of all correct responses (i.e. a word on the response screen was correctly marked as *yes*, i.e. heard, during the 5-word sentence of a trial) divided by the number of all presented words in a condition (0–100%). Effort scores were calculated by averaging a participant's responses on the 5-point Likert scale at the end of each block per condition (1 = low effort, 5 = high effort). Task engagement scores were calculated in the same way as effort scores, and served as a control variable to rule out bias from different task engagements for conditions of selective attention (see Supplementary Statistics).

## Mediation analysis

To investigate whether the top-down control of eye movements could partially contribute to neural differences in selective attention to speech, we conducted an additional analysis based on the logic of a mediation analysis, adapted to our time-resolved regression analyses (i.e. encoding models). For this analysis approach, we favored the boosting algorithm[65] due to how it differently estimates TRFs. While both ridge regression and boosting lead to comparable metrics with regards to accuracy and error[66], boosting begins with a sparsity-prior and allows multiple predictors to compete to explain the dependent variable[63]. In comparison, in order to achieve a similar effect, ridge regression would require the possibility for different regularization parameters for individual models (i.e. plain and direct effect models) or even features (in the sense of banded ridge regression[23,67]). However, this could, by the definition of regularization (i.e. penalization of weights), lead to decreased model weights even in the absence of a mediation effect. Accordingly, we estimated TRFs using the Eelbrain

toolkit[63] using the same time-lags as in the ocular speech tracking analysis from -100 - 550 ms. As the boosting architecture requires continuous input data, we first zero-padded the end of each trial by 700 ms to exclude discontinuation artifacts, concatenated trials within each condition to a continuous segment (similar to ref. [68]), and finally used a 4-fold cross-validation approach based on ℓ2 error minimization to avoid overfitting (topographies of envelope encoding as well as corresponding TRFs validated this approach and ruled out any spurious effects due to zero-padding/trial concatenation, please see Supplementary Figs., Fig. 3). The TRFs that we obtained from our encoding models can be interpreted as time-resolved weights for a predictor variable that aims to explain a dependent variable (very similar to beta-coefficients in classic regression analyses). Based on this assumption, we can try to establish the different contributions in the triad relationship of speech, eye movements, and neural activity (see Fig. 6a). A very well-established finding states that speech acoustics can predict neural activity[69–71]. Given our hypothesis that ocular movements track the speech envelope, we assume this finding to be mediated to some extent by ocular speech tracking. To test this assumption, we simply compared the plain effect of the speech envelope on neural activity to its direct (residual) effect by including an indirect effect via eye movements in our model. Thus the plain effect (i.e. speech envelope predicting neural responses) is represented in the absolute weights (i.e. TRFs) obtained from a simple model:

$$\text{neural response} = \text{TRF(c)}^* \text{speech envelope} \qquad (1)$$

The direct (residual) effect (not mediated by eye movements) is obtained from a model including two predictors:

$$\text{neural response} = \text{TRF(c')}^* \text{speech envelope} + \text{TRF(b)}^* \text{eye movements} \qquad (2)$$

and represented in the exclusive weights (c') of the former predictor (i.e. speech envelope).

Note that the evaluation of the effect of the speech envelope on eye movements (termed "a" in Fig. 6a) preceded this analysis (see previous section). The two models described above were calculated to predict the neural responses of all 102 magnetometer channels separately. Subsequently, we used a cluster-based permutation-dependent t-test to compare TRFs (using absolute values) of both models at each location (note that the polarity of neural responses is not of interest here). If model weights are significantly reduced by the inclusion of eye movements into the model (i.e. c' < c), this indicates that a meaningful part of the relationship between the speech envelope and neural responses was mediated by eye movements (see Fig. 6a). Since no effect of ocular speech tracking was found for the distractor in a single speaker condition (Condition 1), we limited our TRF comparison to the speech envelope encoding of the target speaker in a single speaker (Condition 2), and the target and distractor speaker in a multi speaker context (Condition 3). In order to account for a potential reduction in model weights simply due to the inclusion of an additional predictor or feature scaling, we repeated the same analysis with a shuffled version of eye movements (over time, within conditions) to obtain the 'plain-control' effect ($c_p$), with the rationale that c' < $c_p$. Again, we used a cluster-based permutation-dependent t-test to compare envelope TRFs (using absolute values) of both (plain-control and direct) models at each location (see Supplementary Figs., Fig. 3).

## Statistical analysis
First, we quantified spatial distributions of gaze during stimulation periods to make sure participants were not systematically shifting their gaze away from the Gabor patch during attention to the auditory modality (Conditions 2 and 3). To further investigate spatial differences in gaze distributions across conditions, we contrasted gaze densities of Condition 1, where participants were instructed to focus on the visual modality (i.e. the Gabor patch), with those of Conditions 2 & 3. For this, we performed a cluster-based randomization approach[72], computing the randomization distribution of t-values after 10,000 permutations with a cluster alpha threshold of 0.05 that was then compared against the original contrast at an alpha level of 0.05, Bonferroni corrected. This procedure was carried out for two two-sided contrasts. We report p-values of clusters and t-values alongside degrees of freedom, 95% confidence intervals (CI), and effect size Cohen's d as an average over sensors within a cluster.

To investigate acoustic speech tracking of eye movements under different conditions of attention, we used Bayesian multilevel regression models with Bambi[73], a Python package built on top of the PyMC3 package[74], for probabilistic programming. First, the correlation between predicted eye movements from the combined TRF-models, including control predictors, acoustic features of interest (speech envelope and acoustic onsets), and measured eye movements was calculated (see Fig. 2b). We then subtracted the encoding results (i.e. correlation between predicted and measured eye movements) from a model which included only the control variables to isolate the effect of acoustic tracking from potential confounds. This difference was then averaged over horizontal and vertical channels and used as a dependent variable according to the Wilkinson notation[75]:

$$\text{envelope tracking} \sim 0 + \text{condition} + (1|\text{subject}) \qquad (3)$$

$$\text{acoustic onset tracking} \sim 0 + \text{condition} + (1|\text{subject}) \qquad (4)$$

Note that by removing the Intercept from the model, all conditions have been tested against a zero-effect of tracking.

To directly compare the tracking of the target and distractor speech in the multi-speaker condition, a post hoc model was calculated including only these two encoding results.

To additionally investigate whether ocular speech tracking is related to behavioral performance, we further included intelligibility and subjectively rated listening effort (see TRF Model Estimation and Data Analysis) as dependent variables in separate models. Intelligibility scores were logit-transformed to account for left-skewed data between [0 1]. Independent variables (envelope and acoustic onset tracking) were mean-centered across subjects within the condition before entering the model:

$$\text{intelligibility} \sim \text{condition}^* \text{envelope tracking} + (1|\text{subject}) \qquad (5)$$

$$\text{intelligibility} \sim \text{condition}^* \text{acoustic onset tracking} + (1|\text{subject}) \qquad (6)$$

$$\text{effort} \sim \text{condition}^* \text{envelope tracking} + (1|\text{subject}) \qquad (7)$$

$$\text{effort} \sim \text{condition}^* \text{acoustic onset tracking} + (1|\text{subject}) \qquad (8)$$

Note that intelligibility was only probed for attended speech, therefore only these two conditions (multi vs. single speaker) were included in the behavioral models.

For all models, we used the weakly- or non-informative default priors of Bambi[73] and specified a more robust Student-T response distribution instead of the default Gaussian distribution. To summarize model parameters, we report regression coefficients and the 94% high-density intervals (HDI) of the posterior distribution (the default HDI in Bambi). Given the evidence provided by the data, the prior and the model assumptions, we can conclude from the HDIs that there is a 94% probability that a respective parameter falls within this interval. We considered effects as significantly different from zero if the 94%HDI

did not include zero. Furthermore, we ensured the absence of divergent transitions (r̂ <1.05 for all relevant parameters) and an effective sample size > 400 for all models (an exhaustive summary of Bayesian model diagnostics can be found in ref. [76]).

After establishing ocular speech tracking effects and their relations to behavior, we further quantified the extent to which this tracking and ICA-cleaned neural responses share contributions to speech encoding. Using a mediation analysis approach, we compared model weights of the speech envelope for predicting neural responses with model weights from an encoding model, including eye movements as an additional predictor (see Mediation Analysis). To establish whether there is a significant difference we used a cluster-based randomization approach[72] on all 102 magnetometers, averaging over time lags (from −50 to 500 ms to exclude possible regression edge artifacts). We computed the randomization distribution of $t$-values after 10000 permutations with a cluster alpha threshold of 0.05 that was then compared against the original contrast at an alpha level of 0.05, Bonferroni corrected. This procedure was carried out for three one-sided contrasts (see Fig. 6c), where we compared the plain (c) and direct (residual, c') effects (i.e. absolute TRF weights) for target speech in the single speaker condition (c' < c), for target speech in the multi-speaker condition (c' < c), and finally where speech was the distractor in the multi-speaker condition (c' < c). Subsequently, we report $p$-values of clusters and the effect size Cohen's $d$ as an average over sensors within a cluster.

### Data visualization
Individual plots were generated in python (3.9.12) using matplotlib[77], seaborn[78], and mne-python[79]. Plots were then arranged as cohesive figures with affinity designer (https://affinity.serif.com/en-us/designer/).

### Reporting summary
Further information on research design is available in the Nature Portfolio Reporting Summary linked to this article.

## Data availability
Preprocessed Data required to reproduce the analyses supporting this work are publicly available in the Open Science Framework repository[80] (https://osf.io/m6rfq). Raw data (> 250 GB) will be shared upon request.

## Code availability
Code to analyze preprocessed data and further reproduce results and figures from this manuscript is available at the Open Science Framework repository[80] (https://osf.io/m6rfq).

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

## Acknowledgements

Q.G., J.S., and P.R. are supported by the Austrian Science Fund (FWF; Doctoral College "Imaging the Mind"; W 1233-B). Q.G. and P.R. are also supported by the Austrian Research Promotion Agency (FFG; BRIDGE 1 project "SmartCIs"; 871232) and F.S. is supported by WS Audiology. Thanks to the whole research team. Special thanks to Manfred Seifter for his support in conducting the measurements. Special thanks to Claudia Contadini-Wright for proofreading the manuscript.

## Author contributions

Q.G. and J.S. designed the experiment, analyzed the data, generated the figures, and wrote the manuscript. P.R. and S.R. recruited participants, supported the data analysis, and edited the manuscript. F.S., K.S., T.H., T.P. and M.C. supported the data analysis and edited the manuscript. N.W. designed the experiment, acquired the funding, supervised the project, and edited the manuscript.

## Competing interests

K.S. is an employee of MED-EL GmbH. All remaining authors declare no competing interests.
