## [Peer Review File · Nature Communications]

Eye movements track prioritized auditory features in selective attention to natural speechREVIEWER COMMENTS

Reviewer #1 (Remarks to the Author):

The paper presents an interesting study on how ocular activity may be spontaneously influenced by a dynamic auditory input. Specifically, it is shown that, when the participants attend to speech, their gaze position can follow the speech envelope, intensity fluctuations in speech. Previous studies have barely looked at how ocular activity is influenced by dynamic sound features and I think the findings here are novel. Nevertheless, I have a number of concerns regarding to how to interpret the results.

1. If saccades can synchronize to speech, does the gaze shift rightwards or leftwards per unit change in the speech envelope? Is there any reason why the gaze should shift in the particular direction? Is it related to where the Gabor patch is in the visual field? The gaze has to return after such a shift, is there any hint about when the return occurs? It's important to show the TRF in the main figures for both ocular and neural activity.

2. Related to the last point, I wonder if ocular activity can be better predicted by the envelope if it's characterized by, e.g., the saccade length or some distance measures that ignore the direction information. I also wonder if the blinks/saccades are related to the envelope. Right now, saccades are removed, but it is shown that the gaze position dynamically follows the speech envelope. Is this achieved by microsaccades, smooth pursuit, or the interpolated eye movements during saccades?

3. For the mediation analysis, a potential explanation is that MEG records neural activity that controls envelope-related ocular activity. When eye movements are considered as a model input, such neural activity is regressed out. In contrast, when eye movements are not considered as a model input, such neural activity appears to track the speech envelope and boosts the prediction accuracy (c). For this explanation, it's not that ocular activity mediates envelope-related neural activity, but instead that there exists neural activity that controls envelope-related ocular movement.

4. Please show the ocular and MEG TRFs as the main figures.

Minor issues:

It's not clear whether the sentences and the surrogate sentences are meaningful, and whether they differ in plausibility.

Reviewer #2 (Remarks to the Author):

This manuscript presents research aimed at exploring eye movements during speech listening. The authors record eye tracking data – along with MEG – when participants are 1) being presented with speech but ignoring it while they concentrate on an unrelated visual task, 2) paying attention to a single speech stream while ignoring the visual task, and 3) paying attention to one speech stream while ignoring both a second speech stream and the visual task. The authors model the horizontal and vertical eye movements using temporal response functions – with different acoustic speech features and control features as predictors. They find that eye movements do not track the speech acoustics in condition 1. But eye movements are related to both the speech envelope and acoustic onsets in conditions 2 and 3 – even for distractor speech in condition 3. And they report some significant correlations between this eye movement effect and performance on a behavioral task in condition 2. Finally, the authors test whether or not the eye movements might be contributing to neural signatures of acoustic speech processing. They do this by fitting TRFs between the speech features and the MEG – and then by seeing if these TRFs are different (especially reduced) when eye movements are also included as a predictor. They find the speech-MEG TRFs are reduced in the latter case and conclude

that eye movements contribute to the neural tracking of speech acoustics

Overall I thought this was an interesting and well written manuscript that described a compelling set of results. That said, I had some queries and suggestions for the authors.

Main comments:

1) My main comment on the work – which is very nice – is that it also feels like some opportunities for added value have been missed. In particular, while the focus on the eye movements has produced some cool new findings, it might also have been interesting to see how these eye movements complement other features from the eye tracking data – i.e., blink patterns, saccades, pupil dilation. I know some of these features have been looked at in other work – and you wanted to focus on the new eye movement question. But it still feels like the manuscript might be stronger if the eye movement findings were explored alongside some of these other measures. This is not something that I think would be strictly necessary – I think the manuscript makes a good contribution as it is. So I would just suggest it as something for the authors to consider as a way to potentially strengthen the work further.

2) I found the mediation analysis quite compelling. However, I also wondered about a more direct approach. Specifically, one could model the MEG based on the acoustics – and then model based on acoustics and eye movement data combined. And then assess if the MEG predictions are improved for the combination over the acoustics alone. This seems more direct to me. Interpreting changes in TRF weights can be a little difficult. Although, as I say, I thought the results were compelling. So, again, this is just a suggestion.

3) I thought it might be worth adding a little more detail and discussion on the temporal profile of the eye movement TRFs. Were there any interesting differences in horizontal vs vertical eye movements? Can we interpret the eye movement TRFs based on their temporal profile?

Other comments:

1) I found the discussion section “Gaze and Prioritization of Spectrotemporal Acoustic Information” very interesting. However, a lot of it focused on the links between sounds localization and eye movements – which makes good sense. And I didn’t feeling like the link to your own results – that there is a link between eye movements and non-spatial speech envelope acoustics – really came across all that clearly – although there was some nice discussion later about temporal predictions and visual disengagement. Anyway, I just wanted to suggest that the authors might want to speculate on how their findings might actually relate to eye movements in natural situations when sounds really do come from different locations. That just didn’t really come across to me in that section.

2) I thought that the efforts to link the current findings – because they involve eye movements – to the motor theory of speech was a bit of a stretch. Oculomotor activity is not really what is involved in the motor theory of speech really. So I just thought that that link felt a bit forced.

3) There is a nice PsyArXiv preprint by Fink et al. (2021) that describes modeling of pupil data – including TRFs. It might be of interest to you.

4) I don’t think the behavioral measures is really intelligibility. Intelligibility usually refers to how comprehensible speech is in some kind of background noise. Your measure is a bit more attentional – with a bit of working memory thrown in. Especially in the attended single speaker condition. I am sure that speech is 100% intelligible, but participants vary in their performance based on attention and memory.

5) I wasn’t sure what “prioritized” meant in the title when I first read it. I am still not sure what it means.

Reviewer #3 (Remarks to the Author):

This study describes a novel phenomenon – the “tracking” of speech sounds by eye movements. Gaze positions were found to correlate with the envelope of speech, and this effect partially mediates the

link between speech sound and neural responses. The study incorporates parallel recording of MEG and eye tracking.

The paper is excellently written. The methodology is careful and the analysis thorough (but see my specific comments below). The finding is novel and intriguing. I'm not sure what the phenomenon reflects and what is its mechanistic explanation, and it doesn't seem like the authors have the answers yet (I will comment about this below). But, nevertheless, I think findings like this should be published even when if the interpretation is still only "half-baked". The authors invested a considerable effort in mapping the next essential steps and guiding future research, which I think would be very helpful and increase the impact that this paper could make. To summarize, I believe that this finding could potentially lead to new understandings, and I encourage its publication in Nature Communications, pending a few revisions as I detail below.

1. The phenomenon that the authors describe is quite unusual and I'm not sure I'm convinced by all the possible interpretations that they offer. Overall, the last interpretation that is suggested, of a "gaze aversion" mechanism, seems to me to be most plausible as it combines the temporal and spatial characteristics of the effect. I do think this interpretation could have been examined more closely with a careful analysis of the spatial characteristics of the findings. Do people actually move their gaze away from the Gabor in specific moments when the speech is more relevant? More informative? Where to do they shift their gaze and how is it related to specific timings in the speech input? In general, the analysis seems to me to be lacking in its spatial aspect. It's highly interesting to know not only when people shift their gaze but also where they do so. This aspect is crucial in supporting all the suggested interpretations.

Moreover, I recommend adding referrals to gaze aversion literature. There are quite a few studies on this topic of gaze aversions away from distracting simulation during non-visual tasks (e.g. Glenberg et al. 1998, Doherty-Sneddon 2002, 2005, 2007, Abeles et al. 2017 and more).

2. The first interpretation suggested in the sub-section titled "Gaze and Prioritization of Spectrotemporal Acoustic Information" links this finding to previous studies on the spatial benefits of eye movements in auditory processing. Specifically, the authors review studies on spatial mapping of auditory and visual information and discuss the benefits of spatial selective attention in promoting auditory processing. But it is unclear to me how this can be linked to the present finding without taking into consideration the spatial aspect. As the task was not spatial and the behavior (at least as reported) does not seem to be spatially specific, it is unclear how it can be explained as alignment of spatial maps, or reflect any spatial benefits.

3. I think the link the author make between this effect and reading is intriguing. But here as well, I'm not sure what is the mechanistic explanation supporting this interpretation. The effects of reading may lead to a general shift of gaze toward the reading direction but how could it explain the unique temporal dynamics of this phenomenon (its correlation with the envelope of speech)? I am curious to see this issue being pursued in future studies, but at this stage I would recommend making it clearer that the lateralization finding is post-hoc (assuming that it was, as it's not part of the main analysis) and thus requires replication and validation.

4. The presentation and the analysis are thorough, but I was left with the feeling that I can't estimate how large is the effect. I was wondering if it is possible to graphically demonstrate how this tracking works – to show the signal following the envelope. Something like what is shown on the top left of Figure 2a (but there it's unclear if this is real data and it's very small). Also, I would recommend adding a permutation analysis to show that the effect is larger than what can be expected randomly. The Bayesian statistics provides a similar conclusion, but I think that a permutation analysis would be more convincing and easier to grasp.

5. I found the decision to remove saccades rather strange. Especially since they were then interpolated. It raises the question of why remove them to begin with. They are an important eye movement after all. Removing and interpolating the saccades has a major effect on the temporal

dynamics of the response so it needs to be well justified.

Minor comment: The paper often refers to eye movement as something that can "encode". I think it would be more productive and helpful if it would be clear that eye movements are not neural responses. They don't "encode" things, it's a behavior.

REVIEWER COMMENTS

Reviewer #1 (Remarks to the Author):

The paper presents an interesting study on how ocular activity may be spontaneously influenced by a dynamic auditory input. Specifically, it is shown that, when the participants attend to speech, their gaze position can follow the speech envelope, intensity fluctuations in speech. Previous studies have barely looked at how ocular activity is influenced by dynamic sound features and I think the findings here are novel. Nevertheless, I have a number of concerns regarding to how to interpret the results.

Response:

We would like to thank you for your positive evaluation of our manuscript. We highly appreciate the suggestions and queries and believe they significantly improved our work.

1. If saccades can synchronize to speech, does the gaze shift rightwards or leftwards per unit change in the speech envelope? Is there any reason why the gaze should shift in the particular direction? Is it related to where the Gabor patch is in the visual field? The gaze has to return after such a shift, is there any hint about when the return occurs? It's important to show the TRF in the main figures for both ocular and neural activity.

Response 1.1:

Thank you for pointing out the importance of including the TRFs in the main figures to complete the picture with the spatiotemporal dynamics of ocular speech tracking and further substantiate our interpretations. We included the ocular TRFs for both envelope and acoustic onsets in Figure 2B, and the MEG TRFs as Figure 3B (see below for snippets of the added figure panels and captions, also see response 1.4). We also included additional figures to the supplementary material (in response to 3.4), showing exemplary trials for every participant with the highest correlation between actual eye movements and predicted eye movements by the envelope model. We believe they are also noteworthy with regards to the spatiotemporal dynamics and interpretations thereof. More directly related to your questions, we added the following lines to the results section to help understanding and interpreting the results based on their spatiotemporal profile (ll. 154-169):

“TRFs revealed similar activation patterns for horizontal and vertical eye movements in Conditions 2 & 3 where the auditory modality is attended (see Fig. 2B). For the envelope, we observed positive initial peaks already at ~ 0 lag, with a fairly rapid decrease after ~ 200 ms.

For acoustic onsets, we observed a slower increase to a broader positive peak at ~ 200 ms with a slower decrease compared to the envelope TRFs. In general, TRF weights show a temporally more pronounced pattern for the speech envelope. We did not observe any meaningful weights to speech features when they were ignored in Condition 1. It is important to note that in this case TRFs can be interpreted not only based on their temporal profile but also on their directionality. Eye tracking data delivers positive and negative values on the horizontal and vertical plane (horizontal: + right, - left; vertical: + up, - down) that was preserved by the analysis. Thus, positive TRF peaks can be interpreted as shifts of gaze in the respective direction. However, it is also important to note that TRF weights should be interpreted with great care. For one, our models also included control features that could potentially bias the spatial interpretability of TRFs. For another, the simplistic single trial 5-word structure of the task could bias the temporal interpretability. In both cases, future studies with continuous unisensory designs are required for replication and validation.”

II. 294-296: “[...] This assumption is further supported by neural TRFs (see Fig. 3B), also showing relatively early (pre)activation patterns”

With regards to the gabor patch, we made sure that participants followed the instructions of keeping their gaze on the gabor patch throughout different conditions to avoid this potential bias, hence spatial (top-right) activity patterns did not exceed the boundaries of the gabor. As all reviewers raised questions about interpretations and the necessity to show TRFs, we also addressed these aspects (directionality, timing) more thoroughly throughout the discussion (please also see responses 2.4, 3.1, 3.2, 3.3). However, as mentioned above, we would like to point out that the interpretation of TRFs, similar to classifier weights, should only be interpreted with great care and is limited even further in the presented study due to its short 5-word sentence design and the additional presence of control predictors.

Fig. 2: b The temporal response functions (TRF) for speech envelope and acoustic onsets tracking. TRFs were resampled to 500 Hz for visualization. Shaded areas represent 95% confidence intervals.

Fig. 3. b Exemplary temporal response functions (TRF) for the plain effect (c) and direct (residual) effect (c'). For illustration we chose the channel that showed the highest prediction accuracy for the plain effect (c) in the single speaker target condition ($r_s = 0.12$). TRFs were resampled to 500 Hz for visualization. Shaded areas represent 95% confidence intervals.

2. Related to the last point, I wonder if ocular activity can be better predicted by the envelope if it's characterized by, e.g., the saccade length or some distance measures that ignore the direction information. I also wonder if the blinks/saccades are related to the envelope. Right now, saccades are removed, but it is shown that the gaze position dynamically follows the speech envelope. Is this achieved by microsaccades, smooth pursuit, or the interpolated eye movements during saccades?

Response 1.2:

Thank you very much for your suggestion. We are also highly interested in how other measures of ocular activity could be predicted by the speech envelope, including blinks and saccades (also pupil dilation, but see response to reviewer 2.1). In the current study, we were not able to answer these questions yet simply because of the design, and hence the analysis used: 1. Trial wise stimulation required us to perform a leave-one-trial-out cross validation procedure for the TRF approach. 2. We instructed participants to a) keep their gaze on the gabor patch at the center of the screen and b) try to not blink too excessively during stimulation, i.e. analysis periods to allow for sufficient gaze data. As we state in the manuscript, blink rate was low with $M = 5.00\%$, $SD = 4.27\%$. We additionally calculated the amount of samples that were interpolated by saccades and added this information to the Methods Section in line l. 577 ($M = 1.13\%$, $SD = 2.81\%$). Therefore, many trials did not

include the mentioned features of interest (blinks and saccades), which would invalidate any potential results and hence is an analysis approach that is not even possible to perform on a computational level with the toolbox used (note that the same logic would apply to other algorithms like boosting). This is also one of the reasons why we strongly emphasize the necessity to perform future studies on continuous designs. Studies using audiobooks as stimulus material would allow for this kind of complementary analysis. However, we agree that an approach that ignores the direction of gaze information could be potentially interesting. For this, we reran the analysis on the derivatives of gaze positions that ignore the direction, i.e. velocity, following the same analysis and statistics procedure (regularization parameter estimation etc.). From the results, we can clearly see that our original analysis on 'raw' gaze position data outperformed this approach. Using velocity, Bayesian multilevel models revealed that eye movements only weakly track the envelope of a single speaker when it was presented as the target of attention ($\beta = 0.002$, 94%HDI = [0.001, 0.004]), not when it served as a distractor to the visual modality ($\beta = -0.003$, 94%HDI = [-0.004, -0.001]), nor in the multi speaker condition at all as target ($\beta = 0.001$, 94%HDI = [-0.001, 0.002]) or distractor ($\beta = 0.001$, 94%HDI = [-0.001, 0.002]). We also added a visual illustration of the results below for easier comparison to the main results in the manuscript. Taken together, in the current study, the results could not be achieved by microsaccades (their rate usually ranges between 0.5 - 2 Hz, which would not be enough to cover in our trial design with short speech segments), nor saccades / interpolated eye movements during saccades (only ~ 1% of data points on average). With regards to your question this would favor the idea of smooth pursuit, although from the literature the terminology is not quite accurate either as it refers to a voluntary movement of the eye to track a visual object (Binder, M. D., Hirokawa, N., & Windhorst, U. (Eds.). (2009). Encyclopedia of neuroscience (Vol. 3166). Berlin, Germany: Springer). Another possibility would be slow drift eye movements, however they typically do not exceed $> 0.5^\circ$ of visual angle while we observed stronger ocular movements (also see Supplementary Figures, Fig. 1). We thus referred to the observed behavior simply as gaze.

Fig. in response to 1.2: **The effect of selective attention on speech tracking by velocity of eye movements.** Differences in prediction accuracies (Δr_s) between models that additionally included the speech envelope and a control model for envelope (left panel) and acoustic onsets tracking (right panel) by horizontal eye movements. Statistics were performed using Bayesian regression models. $N = 30$

3. For the mediation analysis, a potential explanation is that MEG records neural activity that controls envelope-related ocular activity. When eye movements are considered as a model input, such neural activity is regressed out. In contrast, when eye movements are not considered as a model input, such neural activity appears to track the speech envelope and boosts the prediction accuracy (c). For this explanation, it's not that ocular activity mediates envelope-related neural activity, but instead that there exists neural activity that controls envelope-related ocular movement.

Response 1.3:

Thank you for your thoughtful feedback. We agree that a causal claim is not warranted with the current regression approach. The mediating effect of eye movements on cortical speech tracking can indeed be bidirectional: the evoked brain activity could follow as a response to sound-related ocular movements, or precede them to control them, or a mixture of both is also possible. The objective of the current work was to raise awareness for this relationship in general and to suggest investigating this phenomenon in more detail in future studies with a more appropriate (i.e. continuous speech) design, in order to target the question of directionality. To point this out more explicitly we changed the wording in lines 196-197 from “we evaluated the influence of eye movements on neural speech tracking with a cluster-based permutation test [...]” to “we evaluated the relationship between eye movements and neural speech tracking with a cluster-based permutation test [...]” and we also added the following statement to the Discussion section (ll. 402-404):

“Also it should be noted that the current findings do not imply a clear directionality - it is possible that the neural activity in question is used to support sound-related ocular activity or vice versa.”

4. Please show the ocular and MEG TRFs as the main figures.

Response 1.4:

We added the ocular TRFs for both horizontal and vertical eye movements to Figure 2B and the MEG TRFs as a new panel in Figure 3B (also see response 1.1). As the encoding approach leads to different TRFs of 102 Magnetometers, we decided to show the MEG TRF for the highest correlating sensor in the single speaker condition ($r_s = 0.12$), using the same sensor for the multi speaker conditions for comparability. We added this information to the figure caption (also see response 1.1).

Minor issues:

It's not clear whether the sentences and the surrogate sentences are meaningful, and whether they differ in plausibility.

Response 1.5:

Thank you for your feedback. We added an example sentence + surrogate sentence as presented in German with an additional translation to English to the Methods Section. We think this gives the reader a better understanding of the stimuli with regards to them being meaningful and plausible. In short, the sentences are meaningful (to the extent a short 5-word sentence by itself can be meaningful without further context of course) and do not differ in plausibility from surrogate sentences:

II. 484-485: “[...] (for example “Peter verleiht vier kleine Bilder.”, in engl.: “Peter lends four small pictures.”).”

II. 488-489: “[...] (for example “Karin bestellt zehn blaue Körbe”, in engl. “Karin orders ten blue baskets”).”

Reviewer #2 (Remarks to the Author):

This manuscript presents research aimed at exploring eye movements during speech listening. The authors record eye tracking data – along with MEG – when participants are 1)

being presented with speech but ignoring it while they concentrate on an unrelated visual task, 2) paying attention to a single speech stream while ignoring the visual task, and 3) paying attention to one speech stream while ignoring both a second speech stream and the visual task. The authors model the horizontal and vertical eye movements using temporal response functions – with different acoustic speech features and control features as predictors. They find that eye movements do not track the speech acoustics in condition 1. But eye movements are related to both the speech envelope and acoustic onsets in conditions 2 and 3 – even for distractor speech in condition 3. And they report some significant correlations between this eye movement effect and performance on a behavioral task in condition 2. Finally, the authors test whether or not the eye movements might be contributing to neural signatures of acoustic speech processing. They do this by fitting TRFs between the speech features and the MEG – and then by seeing if these TRFs are different (especially reduced) when eye movements are also included as a predictor. They find the speech-MEG TRFs are reduced in the latter case and conclude that eye movements contribute to the neural tracking of speech acoustics

Overall I thought this was an interesting and well written manuscript that described a compelling set of results. That said, I had some queries and suggestions for the authors.

Response:

Thank you very much for your positive feedback and highly valuable suggestions. We thoroughly adapted our manuscript with regards to your queries. We firmly believe that they significantly impacted our work towards a favorable evaluation.

Main comments:

1) My main comment on the work – which is very nice – is that it also feels like some opportunities for added value have been missed. In particular, while the focus on the eye movements has produced some cool new findings, it might also have been interesting to see how these eye movements complement other features from the eye tracking data – i.e., blink patterns, saccades, pupil dilation. I know some of these features have been looked at in other work – and you wanted to focus on the new eye movement question. But it still feels like the manuscript might be stronger if the eye movement findings were explored alongside some of these other measures. This is not something that I think would be strictly necessary – I think the manuscript makes a good contribution as it is. So I would just suggest it as something for the authors to consider as a way to potentially strengthen the work further.

Response 2.1:

Thank you very much for your positive response. We agree that there is a huge potential to investigate other features from eye tracking data. That being said, we would like to point out that this added value has been intentionally retained for future studies. With regards to blink patterns and saccades, the single-trial design as well as experiment instructions to a) focus on the gabor patch at the center of the screen and b) to not blink too excessively for valid eye tracking data naturally resulted in many trials without any blinks or saccades (please also see response 1.2). Apart from the many benefits of our design, TRF analysis on trials with 'empty' responses (i.e. no blinks or saccades in a trial) was not possible. With regards to pupil dilation, we also agree that this is a highly interesting measure to take into account. We intentionally retained this kind of analysis for future studies since the rotating gabor sequence on the screen during the stimulation period severely influenced the pupil response. Any dilation measure would thus be so heavily confounded by the visual stimulation that potential responses to speech could not be interpreted with great confidence. This was also another reason why we strongly encourage to replicate and further add value to the experiment in a continuous design without any visual stimulation.

2) I found the mediation analysis quite compelling. However, I also wondered about a more direct approach. Specifically, one could model the MEG based on the acoustics – and then model based on acoustics and eye movement data combined. And then assess if the MEG predictions are improved for the combination over the acoustics alone. This seems more direct to me. Interpreting changes in TRF weights can be a little difficult. Although, as I say, I thought the results were compelling. So, again, this is just a suggestion.

Response 2.2:

Thank you for your encouraging feedback. We chose to compare the TRF weights in order to restrict our interpretation to sound-related neural activity. A model that combines acoustic information and eye movement data will most likely be able to explain more variance in brain activity, and therefore lead to higher prediction accuracies, than a simple model based on acoustics alone. It would, however, be impossible to tell whether this additional activity is in any way related to an improved speech processing. It is reasonable to assume that those ocular movements, which are completely unrelated to the sound, elicit considerable changes in neural activity as well. We therefore decided to focus on the weights of the acoustic predictor. Inspired by the "classic" mediation analysis in standard multiple regression, we have taken advantage of the fact that the mTRF approach is essentially a regression (with the proviso that the weights should only be compared if the regularization parameter is controlled).

3) I thought it might be worth adding a little more detail and discussion on the temporal profile of the eye movement TRFs. Were there any interesting differences in horizontal vs vertical eye movements? Can we interpret the eye movement TRFs based on their temporal profile?

Response 2.3:

Thank you for your valuable input on a more thorough discussion of the temporal profiles. Based on reviewer request, we additionally included the TRFs of ocular and neural data in the main figures now (please also see responses 1.1 and 1.4). We also address the TRFs and how to interpret them in the results section (also see response 1.1):

II. 154-169: “TRFs revealed similar activation patterns for horizontal and vertical eye movements in Conditions 2 & 3 where the auditory modality is attended (see Fig. 2B). For the envelope, we observed positive initial peaks already at ~ 0 lag, with a fairly rapid decrease after ~ 200 ms. For acoustic onsets, we observed a slower increase to a broader positive peak at ~ 200 ms with a slower decrease compared to the envelope TRFs. In general, TRF weights show a temporally more pronounced pattern for the speech envelope. We did not observe any meaningful weights to speech features when they were ignored in Condition 1. It is important to note that in this case TRFs can be interpreted not only based on their temporal profile but also on their directionality. Eye tracking data delivers positive and negative values on the horizontal and vertical plane (horizontal: + right, - left; vertical: + up, - down) that was preserved by the analysis. Thus, positive TRF peaks can be interpreted as shifts of gaze in the respective direction. However, it is also important to note that TRF weights should be interpreted with great care. For one, our models also included control features that could potentially bias the spatial interpretability of TRFs. For another, the simplistic single trial 5-word structure of the task could bias the temporal interpretability. In both cases, future studies with continuous unisensory designs are required for replication and validation”

We further addressed this additional information gain based on TRFs throughout the discussion to substantiate our interpretations with regards to spatiotemporal aspects (please also see responses 2.4, 3.1, 3.2, 3.3)

Other comments:

1) I found the discussion section “Gaze and Prioritization of Spectrotemporal Acoustic Information” very interesting. However, a lot of it focused on the links between sounds localization and eye movements – which makes good sense. And I didn’t feeling like the link to your own results – that there is a link between eye movements and non-spatial speech envelope acoustics – really came across all that clearly – although there was some nice discussion later about temporal predictions and visual disengagement. Anyway, I just wanted to suggest that the authors might want to speculate on how their findings might actually relate to eye movements in natural situations when sounds really do come from different locations. That just didn’t really come across to me in that section.

Response 2.4:

Thank you for highlighting this section of the discussion. In order to comply with yours as well as another reviewer's request (see response 3.2), we toned down this section a bit with regards to the spatial aspect of auditory and visual space maps, pointing out that the current design did use non-spatial stimuli. We thus kept the section of the discussion on prioritization of spectrotemporal acoustic information with more focus on spectrotemporal than spatial characteristics, however including a bit of speculation on natural situations where spatial discriminability is usually a given / important. Yet, we did not push too far here to also comply with the mentioned reviewer request 3.2. We believe the adapted version of this section clarifies a bit of the previous confusion with regards to our own results. More directly related to your request, we would like to highlight the following part that was added in this section of the discussion where we also note that the spatial dynamics of ocular speech tracking could be related to a ‘residual’ mechanism of learned association:

II. 246-260: “The observed effect of ocular speech tracking could reflect the evaluation of attended sound based on spectral features and timing leading to the redirections of gaze that we report as ocular speech tracking. Based on TRFs, we see a systematic tracking of speech with a spatial shift of gaze to a top-rightwards direction within the boundaries of the visual stimulus. It would be intriguing to pursue the spatial dynamics of ocular speech tracking with spatially distributed or even spatially moving speakers. Here, one could also explore saccadic modulations of speech tracking (also on a neural level, see ²⁷) and further detail different eye movements (e.g. (micro)saccades and (micro)saccadic inhibition, slow-drift, smooth pursuit, etc.) and their effects on auditory prioritization and processing. However, with regards to the current design it should be noted that we used a non-spatial task without any meaningful visual stimulation where speech was always presented at phantom center for both single and multi speaker conditions. Ocular speech tracking could

therefore be interpreted as a ‘residual’ activity based on learned associations between the spatio- and spectrotemporal dynamics of speech and respective speakers.”

2) I thought that the efforts to link the current findings – because they involve eye movements – to the motor theory of speech was a bit of a stretch. Oculomotor activity is not really what is involved in the motor theory of speech really. So I just thought that that link felt a bit forced.

Response 2.5:

Thank you for pointing this out. Since we got a lot of very valuable feedback by you and the other reviewers to improve other parts of the discussion, we agree that the link here is in comparison a bit of a stretch and falls a little short with respect to what the data shows. We therefore decided to remove parts that make claims about direct relations to motor theories of speech.

3) There is a nice PsyArXiv preprint by Fink et al. (2021) that describes modeling of pupil data – including TRFs. It might be of interest to you.

Response 2.6:

Thank you very much for your reference to the work of Fink et al. It is indeed a very interesting read. We now also cite the work in the manuscript with reference to dynamic modeling of pupil data as a potential complementary analysis. However, out of the issues mentioned above at 2.1, we stayed away from any analysis of pupil dilation in this work.

l. 369: “[...] for dynamic modeling of pupil data also see ⁵¹)

4) I don’t think the behavioral measures is really intelligibility. Intelligibility usually refers to how comprehensible speech is in some kind of background noise. Your measure is a bit more attentional – with a bit of working memory thrown in. Especially in the attended single speaker condition. I am sure that speech is 100% intelligible, but participants vary in their performance based on attention and memory.

Response 2.7:

Thank you for challenging our view of the behavioral measurement. It is indeed an interesting point to discuss and difficult to argue about. Nonetheless, we tend to disagree that it is not a measurement of intelligibility. If it were more a measurement of working memory and / or attention, we would expect to see a similar response distribution in the multi

speaker condition where the amount of items to memorize (5 words) and the focus of attention (auditory) remain the same. However, we can clearly see a relatively strong decrease in the percentage of correct responses in comparison to the attended single speaker condition with much greater variance. Potentially, one could relate this to task demand, however, we did not find any relations between ocular speech tracking and subjectively perceived listening effort (which we believe would be more closely related to attention and working memory). In addition, in the attended single speaker condition where speech is 100% intelligible, we see a ceiling effect that would most likely show a stronger variance if the measurement entailed more attributes of attention and working memory. In our opinion, the clear increase of response variance from the single speaker to multi speaker condition can thus be mostly attributed to the decrease of intelligibility (due to the presence of a distractor speaker). Also, the speech / sentence material we used for this study is used as a measure of speech intelligibility in clinical settings for ~20 years and was designed and refined to be as “light” (i.e. easy) on working memory and attentional demand as possible to guarantee a correct clinical assessment. We hope you agree with our decision to not change the terminology / cognitive aspect of what the behavioral assessment represents. However, we agree that we cannot entirely rule out the possibility that attentional and working memory contributions differ between conditions and added this potential contribution to the discussion. Also, a potential link of this intelligibility measure to neural quantifications of attention (e.g. alpha oscillations), and working memory (e.g. beta oscillations), or additional behavioral probes could be really intriguing and should be investigated in future studies.

II. 383-386: “In addition, we cannot entirely rule out a potential contribution of individual differences in attention and working memory to the behavioral measure of intelligibility. Future studies should address this possibility in greater detail.”

5) I wasn’t sure what “prioritized” meant in the title when I first read it. I am still not sure what it means.

Response 2.8:

Thank you for pointing this out. In our understanding the term “prioritized” is usually used in the context of (selective) attention, e.g. prioritization of acoustic information. The uncertainty probably stems from the use of “prioritization” *and* “selective attention” at once within the title. We did so with (in our opinion) valid reasoning: 1) We view the design as multisensory with simultaneous auditory (speech) and visual (gabor) stimulation, hence we used “selective attention” to carve out the finding that eye movements track speech when it is attended, but not when it is ignored. 2) We used the word “prioritized” in addition to

“selective attention” to narrow down the phenomenon of ocular speech tracking in that it even differentiates between a target and simultaneously presented distractor, hence “prioritized” features within the auditory modality. Since this was a “minor” comment, we, as of now, kept the terminology of the title. However, if the editors and reviewers agree on this potential pitfall and recommend to change the title, we would of course comply with this request and come up with a different title that avoids the term “prioritized”.

Reviewer #3 (Remarks to the Author):

This study describes a novel phenomenon – the “tracking” of speech sounds by eye movements. Gaze positions were found to correlate with the envelope of speech, and this effect partially mediates the link between speech sound and neural responses. The study incorporates parallel recording of MEG and eye tracking.

The paper is excellently written. The methodology is careful and the analysis thorough (but see my specific comments below). The finding is novel and intriguing. I’m not sure what the phenomenon reflects and what is its mechanistic explanation, and it doesn’t seem like the authors have the answers yet (I will comment about this below). But, nevertheless, I think findings like this should be published even when if the interpretation is still only “half-baked”. The authors invested a considerable effort in mapping the next essential steps and guiding future research, which I think would be very helpful and increase the impact that this paper could make. To summarize, I believe that this finding could potentially lead to new understandings, and I encourage its publication in Nature Communications, pending a few revisions as I detail below.

Response:

We would like to thank you very much for your thorough suggestions and clarifications, especially with regards to the mechanistic explanation. We firmly believe that your suggestions improved the interpretations profoundly and highlight the necessity for careful evaluation of the presented results with replication and validation by future research directions.

1. The phenomenon that the authors describe is quite unusual and I’m not sure I’m convinced by all the possible interpretations that they offer. Overall, the last interpretation that is suggested, of a “gaze aversion” mechanism, seems to me to be most plausible as it combines the temporal and spatial characteristics of the effect. I do think this interpretation could have been examined more closely with a careful analysis of the spatial characteristics

of the findings. Do people actually move their gaze away from the Gabor in specific moments when the speech is more relevant? More informative? Where do they shift their gaze and how is it related to specific timings in the speech input? In general, the analysis seems to me to be lacking in its spatial aspect. It's highly interesting to know not only when people shift their gaze but also where they do so. This aspect is crucial in supporting all the suggested interpretations.

Moreover, I recommend adding referrals to gaze aversion literature. There are quite a few studies on this topic of gaze aversions away from distracting simulation during non-visual tasks (e.g. Glenberg et al. 1998, Doherty-Sneddon 2002, 2005, 2007, Abeles et al. 2017 and more).

Response 3.1:

Thank you for suggesting a more thorough discussion of the spatial characteristics of the effect to further strengthen our interpretations. We would also like to thank you very much for the "gaze aversion" literature. We adapted this section in the discussion, adding more detail on the spatiotemporal characteristics of the effect and how this could be related to gaze aversion in ll. 335-344:

"TRFs show positive weights peaking at ~ 200 ms without any pronounced negative weights within the analyzed time window. Taken together with examples of highest correlating trials for the attended single speaker condition based on the speech envelope (see Supplementary Figures, Fig. 2 & Fig. 3), this indicates a successive shift of gaze in a top-right directionality. In addition, we verified that participants kept their gaze within the boundaries of the distracting visual stimulus throughout the stimulation period. Taken together, the spatial dynamics of ocular speech tracking seem to be too timing specific for a general gaze aversion process. Future studies could directly address the potential principle of gaze aversion by implementing an eyes-closed condition. In this regard, it should be noted that EOG activity seems to align to attended acoustics also in an eyes closed condition¹⁹."

2. The first interpretation suggested in the sub-section titled "Gaze and Prioritization of Spectrotemporal Acoustic Information" links this finding to previous studies on the spatial benefits of eye movements in auditory processing. Specifically, the authors review studies on spatial mapping of auditory and visual information and discuss the benefits of spatial selective attention in promoting auditory processing. But it is unclear to me how this can be linked to the present finding without taking into consideration the spatial aspect. As the task was not spatial and the behavior (at least as reported) does not seem to be spatially specific,

it is unclear how it can be explained as alignment of spatial maps, or reflect any spatial benefits.

Response 3.2:

Thank you for pointing out that we needed to clarify this section of the discussion. Since the phenomenon of ocular speech tracking is novel, we wanted to cover a greater variety of potential explanations in our discussion. We are aiming to relate our results to a broad field of research in order not to exclude any interpretations in advance and to inspire multiple future directions of research. In order to comply with yours as well as another reviewer's request (please see response 2.4), we toned down this section of the discussion with regards to the previously mentioned auditory and visual space maps. We agree that the link between the non-spatial task and spatial interpretations thereof did not come across all that clearly previously. We now adapted this section of the discussion to comply with both reviewer requests, pointing out that a non-spatial task was used and gaze is more related to spectrotemporal dynamics, additionally pointing out that, as of now, we mechanistically explain the observed gaze behavior as a 'residual' activity.

II. 246-260: "The observed effect of ocular speech tracking could reflect the evaluation of attended sound based on spectral features and timing leading to the redirections of gaze that we report as ocular speech tracking. Based on TRFs, we see a systematic spatial tracking of speech with a shift of gaze to a top-rightwards direction within the boundaries of the visual stimulus. It would be intriguing to pursue the spatial dynamics of ocular speech tracking with spatially distributed or even spatially moving speakers. Here, one could also explore saccadic modulations of speech tracking (also on a neural level, see ²⁷) and further detail different eye movements (e.g. (micro)saccades and (micro)saccadic inhibition, slow-drift, smooth pursuit, etc.) and their effects on auditory prioritization and processing. However, with regards to the current design it should be noted that we used a non-spatial task without any meaningful visual stimulation where speech was always presented at phantom center for both single and multi speaker conditions. Ocular speech tracking could therefore be interpreted as a 'residual' activity based on learned associations between the spatio- and spectrotemporal dynamics of speech and respective speakers."

3. I think the link the author make between this effect and reading is intriguing. But here as well, I'm not sure what is the mechanistic explanation supporting this interpretation. The effects of reading may lead to a general shift of gaze toward the reading direction but how could it explain the unique temporal dynamics of this phenomenon (its correlation with the envelope of speech)? I am curious to see this issue being pursued in future studies, but at this

stage I would recommend making it clearer that the lateralization finding is post-hoc (assuming that it was, as it's not part of the main analysis) and thus requires replication and validation.

Response 3.3:

Thank you for your suggestion. As mentioned above, for now we are left with the mechanistic explanation being a 'residual' activity. We agree that we need to clarify the post-hoc lateralization finding and that it requires replication and validation in tailored designs. We added two parts to the discussion for clarification:

II. 3098-311: "TRFs point towards a successive top-rightwards shift of gaze with similar timing aspects compared to reading of ~ 200 ms (also see Supplementary Figures, Fig. 2 & 3, for illustrations of real and predicted eye movements during speech presentation)."

II. 345-348: "In general, the post-hoc lateralization findings of ocular speech tracking during a non-spatial task without any meaningful visual stimulation requires further replication and validation. As of now, the mechanistic explanation of a residual activity based on learned association seems to be the most applicable for the findings in the current study."

4. The presentation and the analysis are thorough, but I was left with the feeling that I can't estimate how large is the effect. I was wondering if it is possible to graphically demonstrate how this tracking works – to show the signal following the envelope. Something like what is shown on the top left of Figure 2a (but there it's unclear if this is real data and it's very small). Also, I would recommend adding a permutation analysis to show that the effect is larger than what can be expected randomly. The Bayesian statistics provides a similar conclusion, but I think that a permutation analysis would be more convincing and easier to grasp.

Response 3.4:

Thank you for this valuable feedback. What Figure 2a depicts is an illustration of an example speech sound and the corresponding envelope, i.e. a graphic illustration to differentiate the panels left and right based on the stimulus feature that was used to get the results in Figure 2a. However, we completely agree that it would be interesting to add larger examples as they could also help to substantiate the interpretations (especially with regards to the spatial aspects mentioned above). We added the two figures below to the supplementary material (since the review correspondence will also be published alongside the manuscript), showing the highest correlating example trials between the actual (green) and predicted (blue) eye

movements for horizontal and vertical eye movements based on the envelope (gray) for every participant. We refer to the supplementary material at respective sections in the discussion (please see responses 3.1 and 3.3).

With regards to a permutation analysis, we understand that frequentist statistics are still the more common and hence easier to grasp approach for the majority of readers, although it could be argued whether it is more convincing than a Bayes approach depending on subjective preference. However, in our case, we could not perform a permutation analysis to compare the effect to what could be expected randomly. The valid approach to generate a random distribution of prediction accuracies to test against would require us to swap the stimulus - response relationships, i.e. pair the envelope of one trial with the eye movements of another, and based on this continue with the same rationale of a leave-one-trial-out cross-validation for training and testing to generate models and calculate prediction accuracies for n-times of permutations. In the current study, we used speech segments of different lengths and thus different lengths of stimuli and responses for which we cannot swap their relationships as it would result in different vector lengths within the same training / test trial(s). This approach would only be feasible on continuous segments that one could chop up in, e.g. 4 segments of the same length. We therefore, among the elegance of the approach for the multilevel modeling with behavioral measures, believe that the Bayes approach was the best solution for this dataset.

Horizontal Eye Movements

Fig. 2: Example trials for horizontal eye movements in the attended single speaker condition. Highest correlating (Spearman's rank, r_s) trials between true / actual eye movement (green) and predicted eye movement (blue) response based on a model with the envelope as stimulus (gray). Each panel represents an individual participant. $N = 30$.

Vertical Eye Movements

Fig. 3: Example trials for vertical eye movements in the attended single speaker condition. Highest correlating (Spearman's rank, r_s) trials between true / actual eye movement (green) and predicted eye movement (blue) response based on a model with the envelope as stimulus (gray). Each panel represents an individual participant. $N = 30$.

5. I found the decision to remove saccades rather strange. Especially since they were then interpolated. It raises the question of why remove them to begin with. They are an important eye movement after all. Removing and interpolating the saccades has a major effect on the temporal dynamics of the response so it needs to be well justified.

Response 3.5:

Thank you for raising this question. We strongly agree that saccades are an important eye movement, and definitely something to explore in other studies. There was, however, a justified reason for removing them in the current study: Due to the task instruction of keeping their gaze at the center of the screen (where the gabor patch was), participants did / should not show much saccadic eye movements as they were supposed to keep their gaze relatively centered to verify comparable visual input across conditions and prevent

systematic gaze shifts away off screen to make the listening task easier. Thus, in this kind of “fixational” task, saccadic movements are rather seen as a violation of task instruction as compared to natural viewing paradigms where they have been recently shown to have a substantial importance for auditory processing (we also cited this article in the manuscript, Leszczynski et al. 2023). For further analysis we thus had, in principle, two ways to deal with saccades: 1) remove trials with saccades which we decided to be the worse choice as we were already on a very low amount of trials / data (i.e. training time) for a valid TRF analysis, or 2) treat them like blinks, i.e. cut out the samples and interpolate them, as for later data scaling during TRF analysis we did not want our training / test set to be biased by a few excessive saccadic eye movements (i.e. ‘outliers’ that bias the analysis). We argue that the dynamics of gaze are better preserved by interpolating than by not removing saccades. That being said, we now additionally calculated the amount of samples that were interpolated by saccades and added this information to the Methods Section in lines l. 577 ($M = 1.13\%$, $SD = 2.81\%$), showing that only a bare minimum of data was actually interpolated due to saccades.

Minor comment: The paper often refers to eye movement as something that can “encode”. I think it would be more productive and helpful if it would be clear that eye movements are not neural responses. They don’t “encode” things, it’s a behavior.

Response 3.6:

Thank you for spotting this issue. We kind of got mixed up here a bit in the terminology due to the method being used and trying to point out the directionality of the analysis (in comparison to the backward decoding approach typically used with neural data, although one could argue that neurons guiding oculomotor behavior “encode”). For clarity we rephrased the respective sections throughout the manuscript to variations using the term “tracking” (e.g. speech is encoded in eye movements -> eye movements track speech)

REVIEWER COMMENTS

Reviewer #1 (Remarks to the Author):

The authors have very well addressed my previous concerns. I now support the paper to be published in Nature Communications. However, I also suggest the authors to briefly discuss why the gaze shifts top-rightwards. It's fine to speculate or to leave the issue to be investigated by future studies.

Reviewer #2 (Remarks to the Author):

Many thanks to the authors for the nice job they have done in responding to my previous comments (and those of the other reviewers).

I have only one minor remaining suggestion – which is to include a sentence or two in the discussion explaining why the study did not investigate blink patterns, saccades, and pupil dilation. The authors' reply has explained this well – but I think it is still missing from the manuscript and readers will wonder about it. It's just a suggestion.

Reviewer #3 (Remarks to the Author):

The authors have made a considerable effort to improve the manuscript. They have answered some of my concerns, but some have not been completely dealt with .

In my first comment I suggested a careful analysis of the spatial characteristics of the findings. In the revision the authors added a description of such properties in the Discussion section (i.e. that gaze is shifted toward the top-rightward direction and within the boundaries of the stimulus). But I found no details of this analysis and the statistics that led to these conclusions (I may have missed it). The interpretations provided still focus very much on the spatial aspects of eye movements (i.e., where people are looking) but they are not supported by proper analysis and statistics. Since the data is available, and apparently also analyzed, I suggest reporting it in the paper.

I asked to see a figure representing the "speech tracking" and the author have provided one that will be placed in the supplementary material. The figure lacks a few details: What are the time values on the x axis? what is the y axis and what is its scale? Also, the blue and green colors are difficult to differentiate. But what concerns me more is that I do not see "tracking" in any of these trials. I see a monotonically rising signal (probably ocular drift) that follows a global monotonic rise of the speech envelope. But there is no tracking of the oscillatory pattern of the envelope itself (the eye tracking signal does not go up and down following the speech envelope). This concerns me because after seeing these images I feel that the description of these findings as reflecting the tracking of speech by eye movements may be a bit misleading. I would be happy to be convinced otherwise.

Finally, I understand now that interpolation was done only for large saccades that were viewed as "task violation". But what about the many microsaccades in the data? Are they analyzed as part of the data? Interpolated? I see that reviewer 1 also asked about microsaccades but from a different perspective. I am interested to know how they were treated in the analysis.

REVIEWER COMMENTS

Reviewer #1 (Remarks to the Author):

The authors have very well addressed my previous concerns. I now support the paper to be published in Nature Communications. However, I also suggest the authors to briefly discuss why the gaze shifts top-rightwards. It's fine to speculate or to leave the issue to be investigated by future studies.

Response 1: We would like to thank you very much for supporting our paper and your highly valuable feedback during the two rounds of revisions. We added your suggestion to the discussion of the manuscript:

ll.256-258: "Future studies are needed to further investigate whether this systematic shift is present also in other designs, especially with continuous speech designs (audiobooks)".

Reviewer #2 (Remarks to the Author):

Many thanks to the authors for the nice job they have done in responding to my previous comments (and those of the other reviewers).

I have only one minor remaining suggestion – which is to include a sentence or two in the discussion explaining why the study did not investigate blink patterns, saccades, and pupil dilation. The authors' reply has explained this well – but I think it is still missing from the manuscript and readers will wonder about it. It's just a suggestion.

Response 2: We would also like to thank you so much for all the great suggestions and contributions to our manuscript. We also appreciate your suggestion to include this additional information in the discussion of the manuscript:

ll. 365-373: "Along these lines, it will be important to investigate how other measures of ocular activity could be predicted by the speech envelope, including blinks, pupil dilation, and (micro)saccades (also see ¹⁹). However, many trials did not include the mentioned features

of interest (blinks and (micro)saccades), which makes any analysis in this direction problematic (along with its interpretation). Additionally, the rotating gabor sequence on the screen during the stimulation period severely influenced the pupil response. Any dilation measure would thus be so heavily confounded by the visual stimulation that potential responses to speech could not be interpreted with great confidence. We therefore strongly encourage replicating and adding further value to this study in a continuous design without any visual stimulation.”.

Reviewer #3 (Remarks to the Author):

The authors have made a considerable effort to improve the manuscript. They have answered some of my concerns, but some have not been completely dealt with . In my first comment I suggested a careful analysis of the spatial characteristics of the findings. In the revision the authors added a description of such properties in the Discussion section (i.e. that gaze is shifted toward the top-rightward direction and within the boundaries of the stimulus). But I found no f details of this analysis and the statistics that led to these conclusions (I may have missed it). The interpretations provided still focus very much on the spatial aspects of eye movements (i.e., where people are looking) but they are not supported by proper analysis and statistics. Since the data is available, and apparently also analyzed, I suggest reporting it in the paper.

I asked to see a figure representing the “speech tracking” and the author have provided one that will be placed in the supplementary material. The figure lacks a few details: What are the time values on the x axis? what is the y axis and what is its scale? Also, the blue and green colors are difficult to differentiate. But what concerns me more is that I do not see “tracking” in any of these trials. I see a monotonically rising signal (probably ocular drift) that follows a global monotonic rise of the speech envelope. But there is no tracking of the oscillatory pattern of the envelope itself (the eye tracking signal does not go up and down following the speech envelope). This concerns me because after seeing these images I feel that the description of these findings as reflecting the tracking of speech by eye movements may be a bit misleading. I would be happy to be convinced otherwise.

Finally, I understand now that interpolation was done only for large saccades that were viewed as “task violation”. But what about the many microsaccades in the data? Are they

analyzed as part of the data? Interpolated? I see that reviewer 1 also asked about microsaccades but from a different perspective. I am interested to know how they were treated in the analysis

Response 3:

Thank you very much for your detailed feedback on our first revision and your further contributions in this round. Your feedback improved our work a lot. Apparently, we misunderstood and missed some of the aspects that were requested in your previous comments. With regards to the spatial characteristics of the findings, we previously referred to a supplementary figure that, in fact, lacked a complete description of the analysis and statistics. We, first of all, reworked this figure including gaze density distributions during stimulus presentations and performed a cluster-based permutation test to contrast the conditions. We added information on analysis and statistics to respective sections in the manuscript. Additionally, we now illustrate (see Fig.3 below) and report this in the main article. Interpretations that focus on the spatial aspects are now referred to these results throughout the manuscript.

ll.555-557: "To further calculate gaze density during trial periods, we followed the same analysis procedure as in ¹⁶. In short, a 2D density histogram was created after multiplying each sample point of gaze on the horizontal and vertical plane with a Gaussian filter."

ll.689-698: "First, we quantified spatial distributions of gaze during stimulation periods to make sure participants were not systematically shifting their gaze away from the gabor patch during attention to the auditory modality (Condition 2 & 3). To further investigate spatial differences in gaze distributions across conditions, we contrasted gaze densities of Condition 1, where participants were instructed to focus on the visual modality (i.e. the gabor patch), with those of Conditions 2 & 3. For this, we performed a cluster-based randomization approach ⁷¹, computing the randomization distribution of t-values after 10000 permutations with a cluster alpha threshold of 0.05 that was then compared against the original contrast at an alpha level of 0.05, Bonferroni corrected. This procedure was carried out for two two-sided contrasts. We report *p*-values of clusters and effect size Cohen's *d* as average over sensors within a cluster."

II.122-131: “First, we investigated spatial gaze characteristics during trials. Importantly, densities confirm that participants kept their gaze on the visual stimulus (i.e. gabor patch) at the center of the screen (see Fig. 3A). Additionally, we contrasted Conditions 2&3 (attend speech) against Condition 1 (attend gabor) using a cluster-based permutation test (see Fig. 3B). For both contrasts, we observed a slight shift of gaze to the top-right whenever the auditory modality is attended. Contrasting Condition 2 (attend single speaker) against Condition 1 revealed one positive cluster ($p < 0.001$, Cohen’s $d = 1.41$), and one negative cluster ($p < 0.001$, Cohen’s $d = -2.07$). Similarly, the cluster-based permutation test revealed one positive cluster ($p < 0.001$, Cohen’s $d = 1.14$), and one negative cluster ($p < 0.01$, Cohen’s $d = -1.59$) for the Contrast Condition 3 (attend a target in a multi speaker condition) vs. Condition 1.”

Fig. 3: Gaze behavior during sound presentations and gabor rotations. **a** Average gaze density for the three conditions (from left to right: Condition 1, Condition 2, Condition 3). Screens are illustrated in degrees of visual angle (dva). Density distributions validate that participants focused their gaze on the visual stimulus throughout all conditions of selective attention. **b** Cluster-based permutation tests on gaze density distributions show a slight shift of gaze to the top-right whenever the auditory modality is attended (Contrast Condition 2 vs 1: positive cluster $p < 0.001$, Cohen’s $d = 1.41$, negative cluster $p < 0.001$, Cohen’s $d = -2.07$. Contrast Condition 3 vs 1: positive cluster $p < 0.001$, Cohen’s $d = 1.14$, negative cluster $p <$

0.01, Cohen's $d = -1.59$). Note that the 'widespread' cluster distributions result from extremely low density values that are not visible in a). $N = 30$.

Additionally, we obviously misunderstood your request on the 'speech tracking' figure. Instead of showing the exemplary trials of the TRF analysis for individual participants (that lacked x-axis details due to different time scales, i.e. trial lengths, and y-axis descriptions due to scalings into the same plot) in the supplementary, we now show exemplary trials of two participants that nicely illustrate how their gaze on the horizontal and vertical plane follows the envelope fluctuations. We completely agree that this is a supportive figure for the findings and especially makes the idea of 'ocular speech tracking' more graspable to the reader. We would like to thank you for pointing this out again and now also show this as part of Figure 2 (see below) in the main article. With regards to a concern of 'slow drift' instead of 'tracking' oscillatory patterns we argue that 1) the chosen time-lags for the TRF analysis of -100 - 550 ms are too short to fit such a slow drift over trial lengths of ~ 2 seconds, 2) the weights illustrated in (now) Figure 4B clearly show a successive 'stepwise' shift pattern instead of a slow-drift. We agree that the Supplementary plots were a bit misleading here previously due to the scaling format for the plots that obscured that stepwise top-rightwards gaze pattern. As mentioned above, we therefore decided to illustrate 'ocular speech tracking' differently by showing the 'pure' gaze and corresponding envelopes scaled into the same range (rescaled between 0-1).

Fig. 2: The approach to establish effects of ocular speech tracking. a Example trials of two participants (S5, S27) show how their measured gaze on the horizontal and vertical plane follows envelope fluctuations (data was rescaled between 0 - 1 for illustration). **b** A regularized linear regression approach called temporal response functions (TRF) was used to predict how features of speech are tracked by eye movements. The difference in prediction accuracy for a control model (C) and combined models that additionally contained the speech envelope (CE) or acoustic onsets (CO) was used to estimate ocular speech tracking solely related to the acoustic features of interest, i.e. speech envelope and acoustic onsets. Prediction accuracies were calculated by Spearman's rank correlation between measured eye movements (mr) and predicted eye movements (pr). **c** We expected ocular speech tracking to be modulated by task induced selective attention. Tracking difference of combined and control models (i.e. pure speech tracking) was expected to be higher whenever sentences were the target in a single speaker or multi speaker condition. For statistical computations we used Bayesian multilevel regression models and illustrated the posterior distributions.

Finally, with regards to microsaccades, we did not exclude them from the data for further analysis. For one, their contribution to the analysis time-windows of trials during speech presentation was low. We ran an analysis on the gaze data with a microsaccade detection

algorithm proposed by Liu, Nobre, & van Ede (2022) and found an amount of $M = 3.70\%$, $SD = 2.00\%$ of samples that showed microsaccadic movements. For another, if they were the driving source of our effects, we would have found significant encoding for velocity of eye movements (which are in principle the key variable of which (micro)saccades are derived from) in the analysis that was requested in the first revision by Reviewer 1 (response 1.2, also see the figure of this response below). In sum, microsaccades were left in the data (if present at all) and not further interpolated. We completely agree with you and Reviewers 1 & 2 that all of the features that eye tracking data delivers, including microsaccades, will be a highly interesting feature to analyze in a continuous audiobook design.

Fig. in response to revision 1, 1.2: **The effect of selective attention on speech tracking by velocity of eye movements.** Differences in prediction accuracies (Δr_s) between models that additionally included the speech envelope and a control model for envelope (left panel) and acoustic onsets tracking (right panel) by horizontal eye movements. Statistics were performed using Bayesian regression models. $N = 30$

References: Liu, B., Nobre, A. C., & van Ede, F. (2022). Functional but not obligatory link between microsaccades and neural modulation by covert spatial attention. *Nature Communications*, 13(1), 3503.

REVIEWER COMMENTS

Reviewer #3 (Remarks to the Author):

The authors have done a great job answering all my concerns and have provided some new convincing evidence. I find this paper ready for publication. Congratulations.

Reviewer #4 (Remarks to the Author):

I have no noteworthy expertise in the fields of communication, eye movements or speech perception. I am well versed in Bayesian hierarchical modeling and was asked to review the Bayesian modeling that is used in this work. Hence, I will only comment on statistical aspects of the manuscript.

The manuscript presents a sophisticated analysis of a rich data set. While I think the general modeling approach is well-founded, there are some shortcomings that should be addressed. I hope my comments are clear, constructive, and help the authors to improve their manuscript. As I am joining the review process at an advanced stage, I apologize if any of the following issues have already been raised and discussed:

1. The indicator of ocular speech tracking, as I understand it, is a difference in rank correlation coefficients. Correlations are calculated between observed eye movements (horizontal and vertical) and eye movements predicted by two models: (1) a model using the onset of visual and auditory stimuli as predictors (control model) and (2) a model using characteristics of the auditory stimulus as predictors (speech envelope or acoustic onset). The difference between the rank correlations for each model is taken as a measure of ocular speech tracking. These rank correlation differences are submitted to hierarchical linear models assuming t-distributed residuals (for greater robustness to outliers compared to normally distributed residuals). My concern with this approach is that this distributional assumption is not well suited for the analysis of correlation coefficients and their differences. A more appropriate procedure would be to Fisher z-transform the rank correlation coefficients before calculating means or differences (doi: <10.1002/9781118445112.stat05964>). This transformation ensures that the dependent variable is approximately normally distributed and unbounded. In addition, in Fisher z-space, differences in rank correlations near the boundaries are weighted differently from those in the center, which is typically considered desirable. Unfortunately, addressing this concern may necessitate a substantial portion of the analyses to be redone.

2. To investigate whether ocular speech tracking is related to adaptive behavior, the authors use linear models to predict their ocular speech tracking index from a measure of intelligibility and subjective effort. This approach is somewhat surprising, as it appears that using intelligibility and subjective effort as dependent variables might align better with the hypothesis being tested. Conceptually, I assume that intelligibility should be influenced by stimulus processing (e.g., ocular speech tracking) rather than the other way around. Perhaps more importantly, this analysis would treat outliers in intelligibility differently. The current approach does not account for measurement error in the predictor variable---only in the dependent variable (as is standard in linear regression models). Looking at Figure 4c, I am concerned that the reported interaction (a notably stronger association between intelligibility and ocular speech tracking in the single-speaker target than in the multi-speaker target condition) may be the result of two or three outliers with relatively low intelligibility. Modeling intelligibility using a t-distribution may mitigate the effect of these outliers. Alternatively, I recommend reporting the results of a robustness analysis that excludes these observations. Finally, I believe a maximal model (doi: <10.1016/j.jml.2012.11.001>) should include random slopes for the main effects of the continuous predictors, as omitting these random slopes may yield overconfident posterior distributions.

3. The Bayesian paradigm is often celebrated for its ability to quantify support for the null hypothesis.

The authors use posterior inference (rather than e.g. model comparison by Bayes factors) and infer effects when the posterior distribution is concentrated away from the null value. Conversely, they infer no effect when the bulk of the posterior distribution is around the null value. My issue with this approach is that it does not allow one to distinguish between absence of evidence and evidence of absence. Within the posterior inference paradigm, evidence of absence requires defining a smallest effect size of interest or a region of practical equivalence around the null value (e.g., doi: <10.1037/met0000402>). I recommend that the authors specify regions of practical equivalence for their dependent variables if they wish to infer the absence of effects.

4. When presenting interaction effects and the results of the mediation analysis, I think it would be useful to also present estimates (and HDI) for the 'simple effects'. While reading the paper I wondered if there was any evidence of a non-zero association between intelligibility and ocular speech tracking in the multi-speaker target condition (slope of the red line in Figure 4c). It is difficult to derive this information from the coefficient values in Table S2 because the coding scheme used for the categorical variables is not mentioned. Similarly, the direct residual temporal response function in the mediation model appears to be flat and located at 0 (at least at the exemplary scalp location used in Figure 5b). A topological map of the residual direct effects (c') might help to interpret these results (in addition to the difference of c and c' , similar to Figure 5c). From my understanding, this would imply that there is no residual direct association between speech envelope and neural response. Assuming that this is a left-parietal location, would this not suggest that neural activity in this region, which is typically associated with speech processing, is fully mediated by eye movements and no speech processing occurs here? Or would this suggest that the reported mediation is indicative of sound-related eye movements (i.e., the reverse causal relationship that the authors touch on in passing in the Discussion, ll. 414-416)? If so, claims such as "we demonstrate that ocular speech envelope tracking contributes to the neural tracking effects of speech over sensors suggestive of auditory processing regions." (ll. 226-228) should be toned down.

REVIEWER COMMENTS

Reviewer #3 (Remarks to the Author):

The authors have done a great job answering all my concerns and have provided some new convincing evidence. I find this paper ready for publication. Congratulations.

Response 3:

We would like to thank you very much for your highly valuable input throughout the review process. It improved the manuscript a lot!

Reviewer #4 (Remarks to the Author):

I have no noteworthy expertise in the fields of communication, eye movements or speech perception. I am well versed in Bayesian hierarchical modeling and was asked to review the Bayesian modeling that is used in this work. Hence, I will only comment on statistical aspects of the manuscript.

The manuscript presents a sophisticated analysis of a rich data set. While I think the general modeling approach is well-founded, there are some shortcomings that should be addressed. I hope my comments are clear, constructive, and help the authors to improve their manuscript. As I am joining the review process at an advanced stage, I apologize if any of the following issues have already been raised and discussed:

1. The indicator of ocular speech tracking, as I understand it, is a difference in rank correlation coefficients. Correlations are calculated between observed eye movements (horizontal and vertical) and eye movements predicted by two models: (1) a model using the onset of visual and auditory stimuli as predictors (control model) and (2) a model using characteristics of the auditory stimulus as predictors (speech envelope or acoustic onset). The difference between the rank correlations for each model is taken as a measure of ocular speech tracking. These rank correlation differences are submitted to hierarchical linear models assuming t-distributed residuals (for greater robustness to outliers compared to normally distributed residuals). My concern with this approach is that this distributional assumption is not well suited for the analysis of correlation coefficients and their differences. A more appropriate procedure would be to Fisher z-transform the rank correlation coefficients before calculating means or differences (doi: <10.1002/9781118445112.stat05964>). This transformation ensures that the dependent variable is approximately normally distributed and unbounded. In addition, in Fisher z-space, differences in rank correlations near the boundaries are weighted differently from those in the center, which is typically considered desirable. Unfortunately, addressing this concern may necessitate a substantial portion of the analyses to be redone.

Response 4.1:

We would like to thank you very much for your suggestion to use Fisher z-transformation on the rank correlations to compute differences and averages. We redid all necessary parts of the analysis and Fisher z-transformed Spearman rank correlations before any further computations (i.e. already during leave-one-trial-out cross-validation for single-trial prediction accuracies). We then used the Fisher z-transformed data for Bayesian modeling. Importantly, using the same formulas

$$\begin{aligned} \text{envelope tracking} &\sim 0 + \text{condition} + (1|\text{subject}) \\ \text{acoustic onset tracking} &\sim 0 + \text{condition} + (1|\text{subject}) \end{aligned}$$

we still observed very similar ocular speech tracking (and also adaptive behavior, see 4.2) effects and conclusions. Table 1 shows the results based on Fisher z-Transformation, Table 2 based on Spearman ranks.

Table 1: Model summary statistics for encoding of acoustic features (Fisher z-transformed ranks) depending on condition

	speech envelope				acoustic onsets			
	b	sd	hdi 3%	hdi 97%	b	sd	hdi 3%	hdi 97%
single speaker - distractor	0.00566	0.00487	-0.00351	0.01474	0.00443	0.00546	-0.00566	0.01504
single speaker - target	0.03175	0.00479	0.02242	0.04055	0.04031	0.00536	0.03002	0.05029
multi speaker - target	0.05963	0.00500	0.05058	0.06924	0.05984	0.00564	0.04917	0.07021
multi speaker - distractor	0.05301	0.00488	0.04358	0.06184	0.04822	0.00559	0.03751	0.05834

Note: Dependent Variable = encoding results: encoding model - control model (average over channels)

Table 2: Model summary statistics for encoding of acoustic features (Spearman ranks) depending on condition

	speech envelope				acoustic onsets			
	b	sd	hdi 3%	hdi 97%	b	sd	hdi 3%	hdi 97%
single speaker - distractor	0.00170	0.00229	-0.00262	0.00597	0.00066	0.00229	-0.00351	0.00514
single speaker - target	0.01066	0.00223	0.00653	0.01481	0.01445	0.00225	0.01003	0.01848
multi speaker - target	0.02630	0.00233	0.02215	0.03090	0.02011	0.00225	0.01566	0.02402
multi speaker - distractor	0.02064	0.00215	0.01664	0.02471	0.01678	0.00229	0.01258	0.02119

Note: Dependent Variable = encoding results: encoding model - control model (average over channels)

Crucially, post-hoc comparison still leads to a ‘significant’ difference for ocular speech envelope tracking in the multi speaker condition since the HDI did not include zero ($\beta = -0.00673$, 94%HDI = $[-0.01342, -0.00004]$) when using Fisher z-Transformed data. In addition - and an improvement to the original analysis with Spearman rank data - we now also observe compelling evidence for an effect for acoustic onset tracking ($\beta = -0.01206$, 94%HDI = $[-0.01950, -0.00464]$) with Fisher z-Transformed data.

In sum, we were able to replicate our previously reported findings of ocular speech tracking for attended speech, with a ‘significant’ difference in the multispeaker condition for both speech features (envelope and acoustic onsets), also with Fisher z-transformed values. We are therefore even more confident in our findings. We consequently adapted the following parts in the manuscript:

1) We now already inform on the Fisher z-transform in the main Figure 2C:

2) We split Figure 4, now it shows the new ocular speech tracking results from Bayes models with Fisher z-transformed ranks in Figure 4A, and the behavioral results in the new Figure 5 (see response 4.2):

3) We adapted the results section (also rephrasing claims about evidence of absence to adhere with 4.3):

II.126-138: “Bayesian multilevel models with Fisher z-transformed prediction accuracies of ocular speech tracking as dependent variables (z') revealed compelling evidence that eye movements only track the envelope of a single speaker when it was presented as the target of attention ($\beta = 0.03175$, 94%HDI = [0.02242, 0.04055]), not when it served as a distractor to the visual modality ($\beta = 0.00566$, 94%HDI = [-0.00351, 0.01474]). We observed a similar effect when using acoustic onsets as a predictor, indicating substantial evidence for ocular tracking of the target ($\beta = 0.04031$, 94%HDI = [0.03002, 0.05029]) but not the distractor sentences ($\beta = 0.00443$, 94%HDI = [-0.00566, 0.01504]). For the multi speaker condition, direct post-hoc comparison between target and distractor speech revealed that speech envelope tracking ($\beta = -0.00673$, 94%HDI = [-0.01342, -0.00004]) was weaker for the distractor speaker compared to the target speaker. The same comparison for acoustic onset tracking points towards a similar effect ($\beta = -0.01206$, 94%HDI = [-0.01950, -0.00464]).”

4) We took into account the extended findings of acoustic onset tracking within the discussion. Importantly, we want to point out that this still leads to the same conclusions and interpretations and therefore only led to minor corrections.

2. To investigate whether ocular speech tracking is related to adaptive behavior, the authors use linear models to predict their ocular speech tracking index from a measure of intelligibility and subjective effort. This approach is somewhat surprising, as it appears that using intelligibility and subjective effort as dependent variables might align better with the hypothesis being tested. Conceptually, I assume that intelligibility should be influenced by stimulus processing (e.g., ocular speech tracking) rather than the other way around. Perhaps more importantly, this analysis would treat outliers in intelligibility differently. The current approach does not account for measurement error in the predictor variable—only in the dependent variable (as is standard in linear regression models). Looking at Figure 4c, I am concerned that the reported interaction (a notably stronger association between intelligibility and ocular speech tracking in the single-speaker target than in the multi-speaker target condition) may be the result of two or three outliers with relatively low intelligibility. Modeling intelligibility using a t-distribution may mitigate the effect of these outliers. Alternatively, I recommend reporting the results of a robustness analysis that excludes these observations. Finally, I believe a maximal model (doi: <10.1016/j.jml.2012.11.001>) should include random slopes for the main effects of the continuous predictors, as omitting these random slopes may yield overconfident posterior distributions.

Response 4.2:

Thank you for challenging our analysis approach on adaptive behavior. We previously went with a model including ocular speech tracking as dependent variable and intelligibility + subjective effort as independent variables as it allowed us to include both behavioral measures into the same model. However, we agree that it might be more intuitive for the reader to reverse this direction in two separate models. We therefore (following the rationale of 4.1) used Fisher z-Transformed ranks (mean-centered) as independent variable and intelligibility / effort as dependent variable:

$$\begin{aligned} \text{intelligibility} &\sim \text{condition} * \text{envelope tracking} + (1|\text{subject}) \\ \text{intelligibility} &\sim \text{condition} * \text{acoustic onset tracking} + (1|\text{subject}) \\ \text{effort} &\sim \text{condition} * \text{envelope tracking} + (1|\text{subject}) \\ \text{effort} &\sim \text{condition} * \text{acoustic onset tracking} + (1|\text{subject}) \end{aligned}$$

With respect to excluding outlier data in the intelligibility distribution, however, we have a different opinion as they are the consequence of a ceiling effect in the (easier) single speaker condition. Excluding these values would leave us simply with even less normally distributed ceiling data, which wasn't desirable for further analysis. In fact, the 'outlier' data rather gave the distribution more meaning since lower intelligibility values were also accompanied by lower ocular speech tracking. However, we agree that these 'distant' data points from the distribution center could be problematic in the dependent variable even with a model using a t-distribution. To address this issue, we transformed the left-skewed intelligibility values (ranging between [0 1]; importantly, no values were either exactly 0 or 1) into logit-space which gave us a much nicer distribution and clearly solved the 'outlier' issue (please see the new Figure 5 within the main manuscript as

illustrated below). Crucially, we found convincing evidence for a positive effect of ocular speech envelope tracking on intelligibility ($\beta = 19.11251$, 94%HDI = [8.85920, 29.22324]) with an interaction effect ($\beta = -13.22355$, 94%HDI = [-25.21237, -0.58644]). Importantly, for acoustic onset tracking, we now also found compelling evidence for a positive effect on intelligibility ($\beta = 11.69498$, 94%HDI = [0.57023, 23.10985]), but with no substantial evidence for an interaction effect ($\beta = -5.38008$, 94%HDI = [-17.82181, 6.20978]).

As for subjective effort as a dependent variable, we were able to confirm our previous results and found neither compelling evidence for an effect of ocular speech envelope tracking on subjective effort ($\beta = -8.04769$, 94%HDI = [-19.25225, 3.66282]) nor an interaction ($\beta = 2.34972$, 94%HDI = [-11.44273, 15.96361]). For acoustic onset tracking, we found no compelling evidence for an effect on effort ($\beta = 4.22670$, 94%HDI = [-15.64272, 7.83369]) nor an interaction ($\beta = 4.88605$, 94%HDI = [-17.31028, 7.41695]).

Fig. 5: Ocular speech tracking and its relation to speech intelligibility and subjective listening effort. Intelligibility was probed only for attended speech. We therefore used intelligibility and subjective listening effort scores for targeted speech in the single and multi speaker context to assess its relation to ocular speech tracking. a Ocular speech tracking (differences in Fisher z-transformed prediction accuracies ($\Delta z'$)) was related to intelligibility with an interaction effect only for the speech envelope. There was no substantial evidence for a relation to subjective effort. b Both ocular speech envelope and acoustic onset tracking predict intelligibility (logit-transformed). The interaction effect for ocular speech envelope tracking indicates that the link between intelligibility and envelope tracking is decreased in a multi speaker condition, while for acoustic onsets the effect seems to be similar in both conditions. (shaded areas represent the 94%HDI, dots represent participants). Statistics were performed using Bayesian regression models. A "*" within posterior distributions depicts a significant difference from zero (i.e. the 94%HDI does not include zero). N = 30.

We additionally addressed your request on a maximal model that includes random slopes for the main effects. For Fisher z-transformed prediction accuracies of ocular speech envelope tracking:

model2: intelligibility ~ condition * envelope tracking + (1 + envelope tracking|subject)

For Fisher z-transformed prediction accuracies of ocular acoustic onsets tracking:

model2: intelligibility ~ condition * acoustic onsets tracking +
(1 + envelope tracking|subject)

We also addressed this for subjective effort as dependent variable:

model2: effort ~ condition * envelope tracking + (1 + envelope tracking|subject)

For acoustic onsets:

model2: effort ~ condition * acoustic onsets tracking +
(1 + acoustic onsets tracking|subject)

In all instances, according to (vastly) overlapping ELPDs, we did not find any substantial differences between the maximal model (model2) and the original 'simple' model (model1) that would justify the use of the more complex model over the simpler one. We therefore kept the original model formulations with less complexity as they increase comparability and interpretability of the effects throughout the manuscript (</doi.org/10.1006/jmps.1999.1283>). Similarly, we briefly wanted to mention that it was not our goal to find a model that maximally explains the underlying data, but rather gives us the best interpretable effects based on a rather simple model.

Taken together, we changed respective parts in the manuscript to the adapted model formulations with adaptive behavior as dependent variables, created the new Figure 5 to illustrate these results and added additional tables (Supplementary Table 2 and 3, see below) to the Supplementary Material to summarize the regression outcomes and added the new results to the main manuscript:

II.164-179: "Fisher z-transformed prediction accuracies (z') of ocular speech tracking were included as the independent variable. We found a positive effect for the encoding of the speech envelope ($\beta = 19.11251$, 94%HDI = [8.85920, 29.22324]) and acoustic onsets ($\beta = 11.69498$, 94%HDI = [0.57023, 23.10985]) on intelligibility indicating that, in the single speaker condition, higher intelligibility is reflected in stronger ocular speech envelope and acoustic onset tracking (see Fig. 5A). While we found a negative interaction with the condition of a multi vs. single speaker ($\beta = -13.22355$, 94%HDI =

[-25.21237, -0.58644]) for ocular speech envelope tracking, there was no substantial evidence for an interaction effect when using acoustic onset tracking as independent variable ($\beta = -5.38008$, 94%HDI = [-17.82181, 6.20978]). This indicates that the link between intelligibility and envelope tracking is decreased in a multi speaker condition, while for acoustic onsets the effect seems to be similar in both conditions (see Fig. 5B). There was no compelling evidence for an effect of neither ocular speech envelope ($\beta = -8.04769$, 94%HDI = [-19.25225, 3.66282]) nor acoustic onset tracking ($\beta = 4.22670$, 94%HDI = [-15.64272, 7.83369]) on subjectively perceived effort. A summary of the statistics can be found in Supplementary Tables, Table 2 and 3.”

Supplementary Table 2

Model summary statistics for intelligibility depending on encoding of acoustic features

	speech envelope				acoustic onsets			
	b	sd	hdi 3%	hdi 97%	b	sd	hdi 3%	hdi 97%
Intercept (single speaker)	4.71761	0.14062	4.45883	4.98430	4.71904	0.15180	4.43055	4.99684
Condition (multi speaker)	-2.33610	0.16900	-2.64686	-2.01496	-2.33834	0.17396	-2.67133	-2.02747
Encoding	19.11251	5.50027	8.85920	29.2232	11.69498	5.96981	0.57023	23.10985
Encoding (multi speaker) x Condition	-13.22355	6.61445	-25.21237	0.58644	-5.38008	6.43128	-17.82181	6.20978

Note: Independent Variable = encoding results: encoding model - control model (average over channels)

Supplementary Table 3

Model summary statistics for subjective effort depending on encoding of acoustic features

	speech envelope				acoustic onsets			
	b	sd	hdi 3%	hdi 97%	b	sd	hdi 3%	hdi 97%
Intercept (single speaker)	2.31823	0.16192	1.99954	2.61297	2.32003	0.15949	2.00677	2.61029
Condition (multi speaker)	1.71252	0.18781	1.36073	2.07069	1.70273	0.18035	1.35386	2.03397
Encoding	-8.04769	6.08016	-19.25225	3.66282	-4.22670	6.23876	-15.64272	7.83369
Encoding (multi speaker) x Condition	2.34972	7.34763	-11.44273	15.96361	-4.88605	6.55999	-17.31028	7.41695

Note: Independent Variable = encoding results: encoding model - control model (average over channels)

3. The Bayesian paradigm is often celebrated for its ability to quantify support for the null hypothesis. The authors use posterior inference (rather than e.g. model comparison by Bayes factors) and infer effects when the posterior distribution is concentrated away from the null value. Conversely, they infer no effect when the bulk of the posterior distribution around the null value. My issue with this approach is that it does not allow one to distinguish between absence of evidence and evidence of absence. Within the posterior inference paradigm, evidence of absence requires defining a smallest effect size of interest or a region of practical equivalence around the null value (e.g., doi: <10.1037/met0000402>). I recommend that the authors specify regions of practical equivalence for their dependent variables if they wish to infer the absence of effects.

Response 4.3:

We thank you for this important point! We intentionally decided against a ROPE approach due to the novelty of our findings and consequently no a-priori knowledge that is strictly necessary for the ROPE. In addition, a 'post hoc' definition of a ROPE should be avoided (<doi.org/10.1177/2515245918771304>; <doi.org/10.1038/s41562-018-0311-x>). We therefore highly recommend that future studies base their approach of a ROPE on the findings presented here and highlight this important aspect in the discussion of the manuscript:

II. 362-364: "Future studies should take advantage of the findings presented here and define a region of practical equivalence (ROPE) to further infer support for any potential null-hypothesis testing".

We actually began to introduce the ROPE concept in a recent preprint (<doi.org/10.1101/2023.06.27.546746>) that also replicated the current study's findings, especially with regards to a mediation analysis, in a continuous design. Furthermore, we stayed away from Bayes Factors as they, strictly speaking, also require informative priors. Finally, to adhere with your important issue regarding absence of evidence and evidence of absence, we rephrased and toned down strong claims about evidence of absence. As our intent was not to infer the absence of effects but rather their presence, we now speak of "no compelling evidence that the effect differs from zero" throughout the manuscript whenever the HDI included zero, e.g.:

II.176-178: "There was no compelling evidence for an effect of neither ocular speech envelope ($\beta = -8.04769$, 94%HDI = [-19.25225, 3.66282]) nor acoustic onset tracking ($\beta = 4.22670$, 94%HDI = [-15.64272, 7.83369]) on subjectively perceived effort."

4. When presenting interaction effects and the results of the mediation analysis, I think it would be useful to also present estimates (and HDI) for the 'simple effects'. While reading the paper I wondered if there was any evidence of a non-zero association between intelligibility and ocular speech tracking in the multi-speaker target condition (slope of the red line in Figure 4c). It is difficult to derive this information from the coefficient values in Table S2 because the coding scheme used for the categorical

variables is not mentioned. Similarly, the direct residual temporal response function in the mediation model appears to be flat and located at 0 (at least at the exemplary scalp location used in Figure 5b). A topological map of the residual direct effects (c') might help to interpret these results (in addition to the difference of c and c' , similar to Figure 5c). From my understanding, this would imply that there is no residual direct association between speech envelope and neural response. Assuming that this is a left-parietal location, would this not suggest that neural activity in this region, which is typically associated with speech processing, is fully mediated by eye movements and no speech processing occurs here? Or would this suggest that the reported mediation is indicative of sound-related eye movements (i.e., the reverse causal relationship that the authors touch on in passing in the Discussion, ll. 414-416)? If so, claims such as "we demonstrate that ocular speech envelope tracking contributes to the neural tracking effects of speech over sensors suggestive of auditory processing regions." (ll. 226-228) should be toned down.

Response 4.4:

Thank you for pointing this out. Since we now model behavioral data differently, we also adapted the supplementary tables accordingly (please see response 4.2). Now the coding scheme becomes apparent and further information on 'simple effects' can be derived more readily. With regards to the mediation analysis on TRFs, we believe it is important to point out here that this represents a multivariate machine learning analysis which slightly differs from a 'classic' regression. In this case any (topographical) interpretation of the residual coefficients (i.e. weights) should be avoided, as the '(non-)flatness' of a TRF on its own gives no indication whether or not it can be used to predict brain activity (which would be the direct effect). It is true that our findings indicate a substantial decrease of the weights in regions that are typically associated with auditory (and speech) processing. However it would not be valid to jump to the conclusion that cortical speech tracking is 'fully mediated' via eye movements. Our intention was to inform the reader that ocular movements seem to contribute to speech tracking effects, which have, so far, been interpreted as purely cortical. The current approach, however, does not allow us to differentiate neural tracking effects from ocular tracking effects (which would be a follow-up question for future studies). Several interpretations are possible (e.g. acoustically evoked ocular movements informing neural tracking; neural tracking leading to selective attention activating orienting eye movement etc). To emphasize that the current approach does not allow us to make any claims about directionality or causality of this relationship, we raised this issue in the discussion:

ll.405-407: "Also it should be noted that the current findings do not imply a clear directionality or causality - it is possible that the neural activity in question is used to support sound-related ocular activity or vice versa."

and further adjusted such claims, highlighting that both directions are, in principle, possible, e.g. the respective header for the results section of the mediation analysis:

II.180-181: "Eye movements and neural activity share contributions to speech tracking."

and also furthermore (among other brief inserts of 'shared' and the like), e.g.:

II.397-399: "we provide evidence for a shared contribution of eye movements and neural activity to speech processing over sensors indicative of auditory areas".

II.418-419: "our results suggest a shared contribution of oculomotor and neural activity to speech processing".

REVIEWERS' COMMENTS

Reviewer #4 (Remarks to the Author):

The authors have made considerable efforts to address the concerns I raised in my review. In my opinion, the manuscript has improved as a result. Where the authors have performed reanalyses, the new results confirm (and slightly strengthen) previously made claims. The authors disagreed with some of my comments on the potentially influential observations in the intelligibility analysis. I will try to address their comments before offering two more minor comments.

For the analysis relating tracking indices to intelligibility, I recommended using intelligibility as a criterion and/or testing the robustness of the results with respect to two participants who appear to have significantly lower levels of intelligibility than the rest of the sample. The authors followed the first recommendation, which partially addressed my concern about the influential observations with low intelligibility (by modeling the result with a t-distribution). I would have liked to explore the effect of these observations further myself, but the data are not publicly available (the OSF repository cited in the manuscript is private; this should be changed). However, I have attempted to reconstruct the data from the plot and performed a very rough analog of the full analysis reported by the authors. This analysis, which should be taken with a big grain of salt, suggests that both the main effect of encoding and the interaction with condition seem to depend on including the two low-intelligence individuals in the analysis (see attached figure). I suggest that the authors test this in the proper analysis and, if confirmed, add these results to the supplement. The authors can explain why they think these observations are meaningful, and readers can then decide for themselves. To further help the reader interpret the results, it would be interesting to see if these individuals also reported low effort (i.e., low task engagement).

Finally, I would like to make two minor comments:

- In Supplementary Table 2, the upper bound for the regression coefficient of the interaction for speech envelope is positive (indicating a nonsignificant result). I think this may be a typo, and the negative sign is missing because a negative upper bound is reported in the main text. Which of these results is correct?
- I think the following is a remaining claim of evidence for absence that should be revised: "This indicates that the link between intelligibility and envelope tracking is decreased in a multi speaker condition, while for acoustic onsets the effect seems to be similar in both conditions (see Fig. 5B)." (ll. 173-175).

Reviewer #4 (Remarks to the Author):

The authors have made considerable efforts to address the concerns I raised in my review. In my opinion, the manuscript has improved as a result. Where the authors have performed reanalyses, the new results confirm (and slightly strengthen) previously made claims. The authors disagreed with some of my comments on the potentially influential observations in the intelligibility analysis. I will try to address their comments before offering two more minor comments.

Response 4.1: We would like to thank you very much for your considerable effort and detail in reviewing our manuscript! Your comments and queries improved the manuscript a lot.

For the analysis relating tracking indices to intelligibility, I recommended using intelligibility as a criterion and/or testing the robustness of the results with respect to two participants who appear to have significantly lower levels of intelligibility than the rest of the sample. The authors followed the first recommendation, which partially addressed my concern about the influential observations with low intelligibility (by modeling the result with a t-distribution). I would have liked to explore the effect of these observations further myself, but the data are not publicly available (the OSF repository cited in the manuscript is private; this should be changed). However, I have attempted to reconstruct the data from the plot and performed a very rough analog of the full analysis reported by the authors. This analysis, which should be taken with a big grain of salt, suggests that both the main effect of encoding and the interaction with condition seem to depend on including the two low-intelligence individuals in the analysis (see attached figure). I suggest that the authors test this in the proper analysis and, if confirmed, add these results to the supplement. The authors can explain why they think these observations are meaningful, and readers can then decide for themselves. To further help the reader interpret the results, it would be interesting to see if these individuals also reported low effort (i.e., low task engagement).

Response 4.2:

Thank you for pointing out that the repository was not publicly available. This was not intentional and of course immediately changed. We apologise for this mistake. The data is now publicly accessible in addition to the code. With regards to the actual analysis, the concern was not only addressed by modelling intelligibility as a dependent variable with a t-distribution but also, crucially, by logit-transformation of the dependent variable. After this transformation, the 'outlier' question was solved by mitigating the ceiling effect (see Figure 5B). In addition to that, the visible data points in the previous version of that figure that were questioned to bias the analysis were not from the same participants. In fact, only one participant (subject_id = 18, also in the source file) showed the lowest score for both conditions (albeit, in this case a score of 92% intelligibility in the single speaker condition can at least be questioned to represent a behavioural outlier). In

addition, especially low-scoring participants showed very high effort values in the multi speaker condition, indicating that their low intelligibility score was not due to disengagement from the task but in fact due to task difficulty (what also makes them meaningful with regards to the underlying distribution). To comply with your request, we nonetheless inspected the behavioural data for 'outliers' based on 1.5*inter quartile range and found two participants (id 7 and 18) to be present in both conditions (note, however, their effort score as previously mentioned as well as the very high intelligibility scores of 94% and 96% for 'outliers' in the single speaker condition, and still 80% intelligibility for subject_id 7 in the multi speaker condition):

Single speaker

Id	Intelligibility (effort)
2	0.967347 (4)
7	0.943000 (3.5)
18	0.922667 (2.5)

Multi speaker

Id	intelligibility
4	0.747000 (4.5)
7	0.806250 (4.5)
18	0.653061 (5)
23	0.838583 (5)

We then reran the modelling for intelligibility based on the dataset that excludes these two participants, again using the updated formula from the previous revision where intelligibility was logit-transformed and envelope tracking represented in fisher-z space:

$$\text{intelligibility} \sim \text{condition} * \text{envelope tracking} + (1|\text{subject})$$

Importantly, we still observed a positive effect for the encoding of the speech envelope ($\beta = 13.40829$, 94%HDI = [2.47948, 23.40420]) as in the main analysis. However, we no longer find evidence for an interaction effect ($\beta = -11.51946$, 94%HDI = [-25.20734, 0.88333]). Since the main finding of an effect of ocular speech tracking on intelligibility held true we followed your advice and decided to not additionally include this confirmation of our results additionally in the supplementary. In addition the response letter will be published alongside the manuscript anyway and as mentioned previously, we argue that these scores are 1) not a problem due to logit-transformation and 2) meaningful with regards to distribution and subjective listening effort.

Finally, I would like to make two minor comments:

- In Supplementary Table 2, the upper bound for the regression coefficient of the interaction for speech envelope is positive (indicating a nonsignificant result). I think this may be a typo, and the negative sign is missing because a negative upper bound is reported in the main text. Which of these results is correct?

Response 4.3:

Thank you very much for spotting this typo! Indeed, the negative sign was missing in the supplementary here (we made sure by rerunning the respective analysis). We now corrected this and report the correct negative value (implying an interaction as stated in the main manuscript).

- I think the following is a remaining claim of evidence for absence that should be revised: "This indicates that the link between intelligibility and envelope tracking is decreased in a multi speaker condition, while for acoustic onsets the effect seems to be similar in both conditions (see Fig. 5B)." (ll. 173-175).

Response 4.4:

Thank you for pointing this out. We agree that this indicates evidence for absence. We simply removed this part ("while for acoustic onsets the effect seems to be similar in both conditions") to avoid this claim. The sentence now simply is: "This indicates that the link between intelligibility and envelope tracking is decreased in a multi speaker condition (see Fig. 5B)."

Adapted mediation analysis with Boosting

Background / Need for Adaption:

In the previous version, we last minutely detected a small, but crucial, indexing mistake in the analysis code related to the mediation analysis presented in the last Figure 6. This problem led to a wrong comparison of mediation model paths, where path c was compared to path b - instead of the correct path c' - to establish the direct effect.

In short, we used regularized ridge regression, as implemented in the mTRF-Toolbox (Crosse et al. 2016, 2021), to estimate envelope encoding weights when predicting held-out neural data recorded with 102 MEG magnetometers. The general approach to establishing a mediation effect of neural speech tracking via eye-movements can be broken down into two pieces: investigating temporal response function (TRF) weights from a model that only includes the speech envelope as a predictor for neural data (i.e. plain model) and test for a decrease of these envelope weights as soon as a mediator, here eye-movements, is included into the model (i.e. direct model). In mediation terms, this refers to subtracting path c' from path c , i.e. $c - c'$ (or $c' < c$ in a statistical analysis with cluster-based permutation test). This snippet from Figure 6 illustrates this:

As we were working on a study (Schubert et al., 2023), that in part aims to show similar findings with regards to ocular speech tracking and mediation of neural data, it became obvious that it was not possible to show the same results when using the regularized ridge regression approach, for which we instead used a different kind of temporal response function technique called Boosting (Brodbeck et al., 2023) in the Schubert et al. study, showing the mediation effect right away. Due to this disparity between the two, I went back to the mediation analysis presented in this paper to check why the mediation effects showed to be so strong although using regularized ridge regression. Here, I unfortunately realized that a wrong indexing in the code led to the contrast of $b < c$, instead of c' . Hence, I compared the weight of the eye-movement predictor with the actual envelope weight of the plain model, which is, obviously false. When using the correct contrast, even more unfortunate, the mediation effect disappeared, just as for the previously mentioned study (Schubert et al., 2023) when using ridge regression instead of boosting.

Why Boosting works and Ridge regression does not:

Both algorithms are versions of time-lagged regression using TRFs. With regards to model metrics, e.g. prediction accuracy and mean squared error, they also lead to comparable results (Kulasingham and Simon, 2023). That being said, the crucial difference refers to how the TRFs are being estimated. In ridge regression, TRF estimation (as a reminder, this is what is being compared in the mediation analysis) starts with a simple least-squares error estimation within a training set. Afterwards this filter, or TRF, is optimized along a range of regularization parameters (λ) that penalize large model weights, i.e. smooth the temporal response function and pull it a bit more towards zero. The desired effect of this regularization is to reduce overfitting and get a filter that better applies to the held out test set. Importantly, this regularization is equally applied to all input features. In our case, the envelope as well as the eye-movement weights are being penalized equally. Hence, in all cases, the influence of the mediator (eye movements) on the independent variable (envelope), becomes overshadowed as both features are treated and become penalized equally. As a result, the comparison between a plain model and direct model using the same λ coefficient will lead to the same envelope weights, and no difference, i.e. mediation effect can be observed.

In contrary, the boosting algorithm begins TRF estimation with a sparsity prior, i.e. the TRFs start at exactly zero. Afterwards, the weights of the features are increased in tiny steps until the model error starts to increase. Most importantly with regards to the mediation analysis, the authors state that “For multiple predictors, the search is performed over all the predictors as well as time lags, essentially **letting the different predictors compete** to explain the dependent variable” (Brodbeck et al., 2023). This crucial point in handling TRF estimation different to ridge regression let us establish the desired mediation effect, and also is why we were able to show the mediation effect in Schubert et al. 2023 fairly easy with Boosting, but not with ridge regression.

Applying Boosting to the current study for the Mediation Analysis:

As previously established, the use of Boosting is able to get a profound estimate of the mediation effect in the current study. We therefore applied the Boosting algorithm with the same time-lags as in the previous ridge regression approach (-100 – 550 ms) and performed the exact same statistics as described in the manuscript. Importantly, now the correct contrast $c' < c$ is performed (this has been double checked by several co-authors). Cluster-based permutation tests still revealed a mediation effect by eye movements for the relationship between the speech envelope and neural responses over left parietal sensors in the single speaker condition (bright purple; $t(29) = -4.40$, $p < 0.001$, Cohen's $d = -0.80$), in the multi speaker condition for a target ($t(29) = -4.90$, $p < 0.001$, Cohen's $d = -0.90$) as well as a distractor speaker (yellow; $t(29) = -4.68$, $p < 0.001$, Cohen's $d = -0.85$; see Figure 6c):

Crucially, the mediation effects remained as hypothesised over left parietal sensors. Therefore, all conclusions and interpretations drawn in the current manuscript remained unchanged.

Please also refer to the preprint (Schubert et al., 2023), that shows that Boosting is also able to show the ocular speech tracking effects (replicating the main findings of the study presented here), and therefore no substantial difference to the ridge regression approach was found other

than the crucial estimate of TRFs for the mediation analysis. In addition, we also added a Supplementary Figure 3, that shows sanity checks of the Boosting analysis in the current study: a) Speech encoding over auditory sensors, b) TRFs of this encoding analysis, and c) a control boosting model that included a shuffled version (across time) of eye-movement data, ensuring that a decrease in model weights in c' was not simply related to the inclusion of an additional predictor, but instead to the meaningful information of eye-movements over time (please also see Supplementary material for further information and illustration).

The manuscript including the erroneous analysis and results can be found at <https://www.biorxiv.org/content/10.1101/2023.01.23.525171v1.abstract> in its original form for comparison.

To close, I am again terribly sorry that this mistake in the original code happened, but I hope you will agree on the suitability of the new analysis approach and the fact that neither hypotheses nor conclusions had to be changed.

Kindest regards,

Quirin Gehmacher (also on behalf of all Co-Authors)

References:

Crosse, M. J., Di Liberto, G. M., Bednar, A., & Lalor, E. C. (2016). The multivariate temporal response function (mTRF) toolbox: a MATLAB toolbox for relating neural signals to continuous stimuli. *Frontiers in human neuroscience*, *10*, 604.

Crosse, M. J., Zuk, N. J., Di Liberto, G. M., Nidiffer, A. R., Molholm, S., & Lalor, E. C. (2021). Linear modeling of neurophysiological responses to speech and other continuous stimuli: methodological considerations for applied research. *Frontiers in neuroscience*, *15*, 705621.

Schubert, J., Gehmacher, Q., Schmidt, F., Hartmann, T., & Weisz, N. (2023). Prediction tendency, eye movements, and attention in a unified framework of neural speech tracking. *bioRxiv*, 2023-06.

Brodbeck, C., Das, P., Gillis, M., Kulasingham, J. P., Bhattasali, S., Gaston, P., ... & Simon, J. Z. (2023). Eelbrain, a Python toolkit for time-continuous analysis with temporal response functions. *Elife*, *12*, e85012.

Kulasingham, J. P., & Simon, J. Z. (2022). Algorithms for estimating time-locked neural response components in cortical processing of continuous speech. *IEEE Transactions on Biomedical Engineering*, *70*(1), 88-96.

REVIEWERS' COMMENTS

Reviewer #4 (Remarks to the Author):

Errors of this nature are not uncommon, particularly in intricate analyses as reported in this manuscript. The authors have adeptly addressed the issue, and the justification for opting for Boosting over Ridge Regression appears well-founded in my assessment. No additional comments are warranted.